# A reproducible experimental survey on biomedical sentence similarity: A string-based method sets the state of the art

**Alicia Lara-Clares** [ID]*, **Juan J. Lastra-Díaz** [ID], **Ana Garcia-Serrano** [ID]

NLP & IR Research Group, E.T.S.I. Informática, Universidad Nacional de Educación a Distancia (UNED), Madrid, Spain

* alara@lsi.uned.es

## Abstract

This registered report introduces the largest, and for the first time, reproducible experimental survey on biomedical sentence similarity with the following aims: (1) to elucidate the state of the art of the problem; (2) to solve some reproducibility problems preventing the evaluation of most current methods; (3) to evaluate several unexplored sentence similarity methods; (4) to evaluate for the first time an unexplored benchmark, called Corpus-Transcriptional-Regulation (CTR); (5) to carry out a study on the impact of the pre-processing stages and Named Entity Recognition (NER) tools on the performance of the sentence similarity methods; and finally, (6) to bridge the lack of software and data reproducibility resources for methods and experiments in this line of research. Our reproducible experimental survey is based on a single software platform, which is provided with a detailed reproducibility protocol and dataset as supplementary material to allow the exact replication of all our experiments and results. In addition, we introduce a new aggregated string-based sentence similarity method, called LiBlock, together with eight variants of current ontology-based methods, and a new pre-trained word embedding model trained on the full-text articles in the PMC-BioC corpus. Our experiments show that our novel string-based measure establishes the new state of the art in sentence similarity analysis in the biomedical domain and significantly outperforms all the methods evaluated herein, with the only exception of one ontology-based method. Likewise, our experiments confirm that the pre-processing stages, and the choice of the NER tool for ontology-based methods, have a very significant impact on the performance of the sentence similarity methods. We also detail some drawbacks and limitations of current methods, and highlight the need to refine the current benchmarks. Finally, a notable finding is that our new string-based method significantly outperforms all state-of-the-art Machine Learning (ML) models evaluated herein.

## Introduction

Measuring semantic similarity between sentences is an important task in the fields of Natural Language Processing (NLP), Information Retrieval (IR), and biomedical text mining, among

**Data Availability Statement:** All the necessary information for reproducing the experiments is available in the Spanish Dataverse Network permanent repositories [1] and [2]. The pre-installed software for executing the experiments is published as a Docker image in DockerHub: https://hub.docker.com/r/alicialara/hesml_v2r1, which is also publicly available in [1]. The DockerHub repository is offered as an alternative to obtain the pre-installed software, but it is not mandatory for reproducing the results reported in this work. The source code of HESML-V2R1 is also published in Github: https://github.com/jjlastra/HESML/releases/tag/Release_HESML_V2R1. We have also created a permanent branch with all the necessary code for reproducing the results (https://github.com/jjlastra/HESML/tree/HESML_STS_paper_experiments). Also, these files have been uploaded to the permanent repository [1]. Finally, the protocol for setting up and running our experiments is published on [3] https://www.protocols.io/view/a-reproducibility-protocol-and-dataset-on-the-biomb5ckq2uw. [1] Lara-Clares, Alicia; Lastra-Díaz, Juan J.; Garcia-Serrano, Ana, 2021, "Reproducible experiments on word and sentence similarity measures for the biomedical domain", https://doi.org/10.21950/EPNXTR, e-cienciaDatos, V2 [2] Lara-Clares A, Lastra-Díaz JJ, Garcia-Serrano A., 2022, "HESML V2R1 Java software library of semantic similarity measures for the biomedical domain", https://doi.org/10.21950/AQLSMV, e-cienciaDatos, V2 [3] Lara-Clares A, Lastra-Díaz JJ, Garcia-Serrano A. A reproducibility protocol and dataset on the biomedical sentence similarity; 2022. Protocols.io, v1. https://www.protocols.io/view/a-reproducibility-protocol-and-dataset-on-the-biom-b5ckq2uw. Special licensing restrictions: We provide a reproducibility dataset that contains all the source code, pre-trained models, pre-installed software, and raw input and output data files together with a reproducibility protocol published at protocols.io. However, to reproduce the experiments evaluated in this work, it is necessary to obtain a license from the National Library of Medicine (NLM) of the United States, which allows the use of the UMLS Metathesaurus databases, as well as SNOMED-CT and MeSH ontologies and the NER tools. For this reason, any reader has full access to all our software and data, but they should go to the NLM license page (https://uts.nlm.nih.gov/license.html) to obtain a license to use the software and data resources mentioned above. Once the readers have obtained the license from the NLM, they should write to eciencia@consorciomadrono.es to obtain the password to decrypt the file, or they can download all the required libraries from the NLM website and

others. For instance, the estimation of the degree of semantic similarity between sentences is used in text classification [1–3], question answering [4, 5], evidence sentence retrieval to extract biological expression language statements [6, 7], biomedical document labeling [8], biomedical event extraction [9], named entity recognition [10], evidence-based medicine [11, 12], biomedical document clustering [13], prediction of adverse drug reactions [14], entity linking [15], document summarization [16, 17] and sentence-driven search of biomedical literature [18], among other applications. In the question answering task, Sarrouti and El Alaomi [4] build a ranking of plausible answers by computing the similarity scores between each biomedical question and the candidate sentences extracted from a knowledge corpus. Allot et al. [18] introduce a system to retrieve the most similar sentences in the BioC biomedical corpus [19] called Litsense [18], which is based on the comparison of the user query with all sentences in the aforementioned corpus. Likewise, the relevance of the research in this area is endorsed by the proposal of recent conference series, such as SemEval [20–25] and BioCreative/OHNLP [26], and studies based on sentence similarity measures, such as the work of Aliguliyev [16] in automatic document summarization, which shows that the performance of these applications depends significantly on the sentence similarity measures used.

The aim of any semantic similarity method is to estimate the degree of similarity between two textual semantic units as perceived by a human being, such as words, phrases, sentences, short texts, or documents. Unlike sentences from the language in general use whose vocabulary and syntax is limited both in extension and complexity, most sentences in the biomedical domain are comprised of a huge specialized vocabulary made up of all sorts of biological and clinical terms, in addition to innumerable acronyms, which are combined in complex lexical and syntactical forms.

Currently, there are several papers in the literature that experimentally evaluate multiple methods on biomedical sentence similarity. However, they are either theoretical or have a limited scope and cannot be reproduced. For instance, Kalyan et al. [27], Khattak et al. [28], and Alsentzer et al. [29] introduce theoretical surveys on biomedical word and sentence embeddings with a limited scope. On the other hand, the experimental surveys introduced by Sogancioglu et al. [30], Blagec et al. [31], Peng et al. [32], and Chen et al. [33] among other authors, cannot be reproduced because of the lack of source code and data to replicate both methods and experiments, or the lack of a detailed definition of their experimental setups. For instance, Sogancioglu et al. [30] provide the BIOSSES evaluation dataset evaluated in this work, as well as a Demo application and the source code used in their biomedical sentence similarity dataset (https://tabilab.cmpe.boun.edu.tr/BIOSSES/About.html); however, they do provide neither the MetaMap [34] annotation tool and UMLS ontology subsets MeSH [35] and OMIM [36] versions to reproduce the ontology-based measures nor the Open Access Subset of PubMed Central (http://www.ncbi.nlm.nih.gov/pmc/) dataset used in their training stage. Blagec et al. [31] introduce a comprehensive experimental survey for biomedical sentence similarity measures, providing the detailed hyper-parameters used for training the models, as well as several code and data to allow the training and evaluation of their methods (https://github.com/kathrinblagec/neural-sentence-embedding-models-for-biomedical-applications); however, they provide neither the post-processed biomedical dataset used in their training phase, nor the pre-trained models. Peng et al. [32] provide the pre-trained models and pre-processed dataset used to train the models (https://github.com/ncbi-nlp/BLUE_Benchmark), but they do not provide detailed information about the pre-processing of the dataset. Finally, Chen et al. [33] provide the pre-trained models (https://github.com/ncbi-nlp/BioSentVec) but provide neither the detailed information about the data used for training the models nor the information on the pre-processing stage. Therefore, it is not possible to evaluate their results in our experiments. Likewise, there are other recent works whose results need to be confirmed. For

add all the external libraries manually. Likewise, they should obtain and sign a Data User Agreement from the Mayo Clinic (https://n2c2.dbmi.hms. harvard.edu/data-use-agreement) to use the MedSTS dataset by sending the Data User Agreement form to the authors of MedSTS.

**Funding:** This work was partially supported by the UNED predoctoral grant started in April 2019 (BICI N7, November 19th, 2018) and the CLARA-HD (PID2020-116001RB-C32) project. The funders had no role in study design, data collection and analysis, decision to publish, or preparation of the manuscript. There was no additional external funding received for this study."

**Competing interests:** NO authors have competing interests.

instance, Tawfik and Spruit [37] experimentally evaluate a set of pre-trained language models, whilst Chen et al. [38] propose a system to study the impact of a set of similarity measures on a Deep Learning ensemble model, which is based on a Random Forest model [39].

The main aim of this work is to introduce a comprehensive and very detailed reproducible experimental survey of methods on biomedical sentence similarity to elucidate the state of the problem by implementing our previous registered report protocol [40]. Our experiments are based on our software implementation and evaluation of all methods analyzed herein into a common and new software platform based on an extension of the Half-Edge Semantic Measures Library (HESML) [41, 42], called HESML (http://hesml.lsi.uned.es) for Semantic Textual Similarity (HESML-STS). All our experiments have been recorded into a Docker virtualization image that is provided as supplementary material together with our software [43] and a detailed reproducibility protocol [44] and dataset [43] to allow the easy replication of all our methods, experiments, and results. This work is based on our previous experience developing reproducible research in a series of publications in the area, such as the experimental surveys on word similarity introduced in [45–48], whose reproducibility protocols and datasets [49, 50] are detailed and independently confirmed in two companion reproducible papers [41, 51], and a reproducible benchmark on semantic measures libraries for the biomedical domain [42]. Finally, we refer the reader to our previous work [40] for a very detailed review of the literature on sentence similarity measures, which is omitted here because of the lack of space and to avoid repetition.

## Main motivations and research questions

Our main motivation is the lack of a comprehensive and reproducible experimental survey on biomedical sentence similarity that allows state of the problem to be set out in a sound and reproducible way, as detailed in our previous registered report protocol [40]. Our main research questions are as follows:

**RQ1** Which methods get the best results on biomedical sentence similarity?

**RQ2** Is there a statistically significant difference between the best-performing methods and the remaining ones?

**RQ3** What is the impact of the biomedical Named Entity Recognition (NER) tools on the performance of the methods on biomedical sentence similarity?

**RQ4** What is the impact of the pre-processing stage on the performance of the methods on biomedical sentence similarity?

**RQ5** What are the main drawbacks and limitations of current methods on biomedical sentence similarity?

A second motivation is implementing a set of unexplored methods based on adaptations from other methods proposed for the general language domain. A third motivation is the evaluation in the same software platform of the three known benchmarks on biomedical sentence similarity reported in the literature as follows: the Biomedical Semantic Similarity Estimation System (BIOSSES) [30] and Medical Semantic Textual Similarity (MedSTS) [52] datasets, as well as the evaluation for the first time of the Microbial Transcriptional Regulation (CTR) [53] dataset in a sentence similarity task, despite it having been previously evaluated in other related tasks, such as the curation of gene expressions from scientific publications [54]. A fourth motivation is a study on the impact of the pre-processing stage and NER tools on the performance of the sentence similarity methods, such as that done by Gerlach et al. [55] for stop-words in a

topic modeling task. And finally, our fifth motivation is the lack of reproducibility software and data resources on this task, which allow an easy replication and confirmation of previous methods, experiments, and results in this line of research, as well as encouraging the development and evaluation of new sentence similarity methods.

## Definition of the problem and contributions

The two main research problems tackled in this work are the design and implementation of a large and reproducible experimental survey on sentence similarity measures for the biomedical domain, and the evaluation of a set of unexplored methods based on adaptations from previous methods used in the general language domain. Our main contributions are as follows: (1) the largest, and for the first time, reproducible experimental survey on biomedical sentence similarity; (2) the first collection of self-contained and reproducible benchmarks on biomedical sentence similarity; (3) the evaluation of a set of previously unexplored methods, such as a new string-based sentence similarity method, based on Li et al. [56] and Block distance [57], eight variants of the current ontology-based methods from the literature based on the work of Sogancioglu et al. [30], and a new pre-trained Word Embedding (WE) model based on FastText [58] and trained on the full-text of articles in the PMC-BioC corpus [19]; (4) the evaluation for the first time of an unexplored benchmark, called CTR [53]; (5) the study on the impact of the pre-processing stage and Named Entity Recognition (NER) tools on the performance of the sentence similarity methods; (6) the integration for the first time of most sentence similarity methods for the biomedical domain into the same software library, called HESML-STS, which is available both on Github (https://github.com/jjlastra/HESML) and in a reproducible dataset [43]; (7) a detailed reproducibility protocol together with a collection of software tools and datasets provided as supplementary material to allow the exact replication of all our experiments and results; and finally, (8) an analysis of the drawbacks and limitations of the current state-of-the-art methods.

The rest of the paper is structured as follows. First, we introduce a collection of new sentence similarity methods evaluated here for the first time. Next, we describe a detailed experimental setup for our experiments on biomedical sentence similarity and introduce our experimental results. Then, we discuss our results and answer the research questions detailed above. Subsequently, we introduce our conclusions and future work. Finally, we introduce three appendices with supplementary material as follows. S1 Appendix introduces all statistical significance results of our experiments, whilst S2 Appendix introduces all data tables reporting the performance of all methods with all pre-processing configurations evaluated herein, and the S3 Appendix introduces a reproducibility protocol detailing a set of step-by-step instructions to allow the exact replication of all our experiments, which is published at protocols.io [44].

## The new sentence similarity methods

This section introduces a new string-based sentence similarity method based on the aggregation of the Li et al. [56] similarity and Block distance [57] measures, called LiBlock, as well as eight new variants of the ontology-based methods proposed by Sogancioglu et al. [30], and a new pre-trained word embedding model based on FastText [58] and trained on the full-text of the articles in the PMC-BioC corpus [19].

### The new LiBlock string-based method

Two key advantages of the family of string-based methods are as follows. Firstly, they can be very efficiently computed because they do not require the use of external knowledge or pre-

trained models, and secondly, they obtain competitive results as shown in Table 8. However, the string-based methods do not capture the semantics of the words in the sentence, which prevent them from recognizing semantic relationships between words, such as synonymy and meronymy among others. In contrast, the family of ontology-based methods capture the semantic relationships between words in a sentence pair and obtain state-of-the-art results in the sentence similarity task for the biomedical domain, as shown in Table 8. However, the effectiveness of ontology-based methods depends on the lexical coverage of the ontologies and the ability to recognize automatically the underlying concepts in sentences by using Named Entity Recognition (NER) and Word Sense Desambiguation (WSD) tools, whose coverage and performance could be limited in several application domains. In fact, the NER task is still an open problem [59] in the biomedical domain because of the vast biomedical vocabulary and the complex lexical and syntactic forms found in the biomedical literature. In comparison, the methods based on pre-trained word embedding models provide a broader lexical coverage than the ontology-based ones and obtain better results. However, the methods based on word embedding do not significantly outperform all ontology-based measures in a word similarity task [48] in addition to requiring a large corpus for training, a complex training phase, and more computational resources than the families of string-based and ontology-based methods.

To overcome the drawbacks and limitations of the string-based and ontology-based methods detailed above, we propose here a new aggregated string-based measure called LiBlock and denoted by $sim_{LiBk}$ henceforth, which is based on the combination of a similarity measure derived from the Block Distance [57] and an adaptation from the ontology-based similarity measure introduced by Li et al. [56] that removes the use of ontologies, such as WordNet [60] or Systematized Nomenclature of Medicine Clinical Terms (SNOMED-CT) [61]. The LiBlock similarity measure obtains the best results in combination with the cTAKES NER tool [62], which allows the detection of synonyms of CUI concepts. Nevertheless, the LiBlock method obtains competitive results regarding the state-of-the-art methods with no use, either implicitly or explicitly, of an ontology, as detailed in Table 12.

The $sim_{LiBk}$ method detailed in Eq (1) is defined by the linear aggregation of an adaptation of the Li et al. [56] measure, called $sim_{LiAd}$ (Eq (3)), and a similarity measure derived from the Block Distance measure [57], called $sim_{Bk}$ (Eq (2)). Let be $L_\Sigma$ the set of word sequences in a universal unseen alphabet $\Sigma$, the $sim_{LiBk}$ function returns a value between 0 and 1 which indicates the similarity score between two input sentences, as defined in Eq 1. The $sim_{Bk}$ function is based on the computation of the word frequencies $fr(w_i, s_j)$ for each input sentence $s_1$ and $s_2$ and their concatenation $s_1 + s_2$, as detailed in equation (Eq (2)). The auxiliary function $fr(w_i, s_j)$ returns the frequency of a word $w_i$ in the word sequence $s_j$, whilst the function $fr(w_i, s_1 + s_2)$ returns the number of occurrences of the word $w_i$ in the concatenation of the two word sequences, denoted by $s_1 + s_2$. On the other hand, the $sim_{LiAd}$ function takes two word sets obtained by invoking the $\sigma$ function (Eq (5)) with the sentences $s_1$ and $s_2$, and then it computes the cosine similarity of the two binary semantic vectors corresponding to invoke the $\varphi(S_1)$ function (Eq (4)) with the $\sigma(s_1)$ and $\sigma(s_2)$ word sets. Finally, the $sim_{LiBk}$ score is defined by either the linear combination of $sim_{Bk}$ and $sim_{LiAd}$, as detailed in Eq (1), or $sim_{Bk}$ if $sim_{LiAd}$ is 0.

**A walk-through example.** Algorithm 1 details the step-by-step procedure to compute the $sim_{LiBk}$ function, whilst Fig 1 shows the pipeline for calculating the LiBlock similarity score defined in Eq 1, as well as an example for illustrating an end-to-end calculation of the $sim_{LiBk}$ similarity score of two sentences.

**Algorithm 1** LiBlock sentence similarity measure for two input pre-processed sentences.

```
1: function: simLiBlock (s₁, s₂)              ▷ being s₁, s₂ word
   sequences ∈ L_Σ
2:    S₁ ← σ(s₁)              ▷ word set sentence 1
```

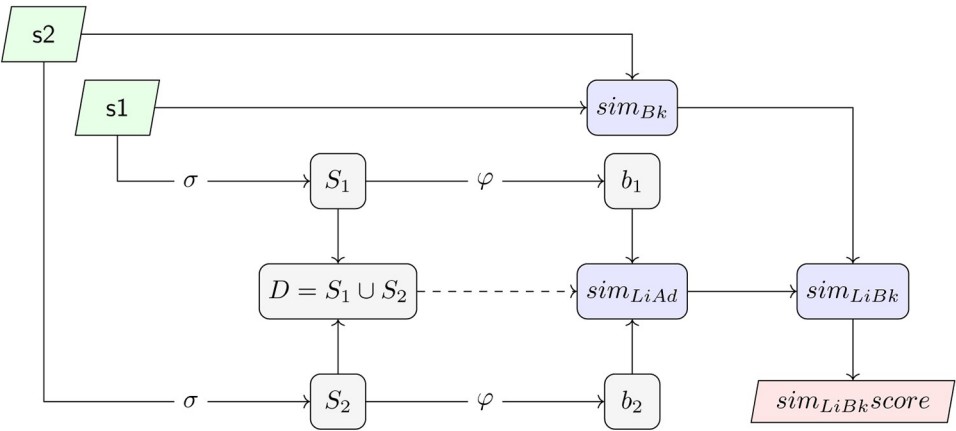

Input : Raw $s_1 \leftarrow$ "Lung tumour formation in mice by oncogenic KRAS requires formation Craf, but not Braf."

Raw $s_2 \leftarrow$ "The oncogenic activity of mutant Kras appears dependent" on functional Craf but not on Braf."

step 1: $s_1 \leftarrow$ {c0280089, formation, mice, oncogenic, c1537502, requires, formation, craf, c0812241}

$s_2 \leftarrow$ {oncogenic, activity, mutant, c1537502, appears, dependent, functional, craf, c0812241}

step 2: $S_1 \leftarrow$ {c0280089, formation, mice, oncogenic, c1537502, requires, craf, c0812241}

step 3: $S_2 \leftarrow$ {oncogenic, activity, mutant, c1537502, appears, dependent, functional, craf, c0812241}

step 4: $D \leftarrow$ {c0280089, formation, mice, oncogenic, c1537502, requires, craf, c0812241, activity, mutant, appears, dependent, functional}

step 5: $b_1 \leftarrow$ {1, 1, 1, 1, 1, 1, 1, 1, 0, 0, 0, 0, 0}

step 6: $b_2 \leftarrow$ {0, 0, 0, 1, 1, 0, 1, 1, 1, 1, 1, 1, 1}

step 7: $sim_{LiAd} \leftarrow 0.471$

step 8: $sim_{Bk} \leftarrow 0.444$

step 9: $sim_{LiBk} \leftarrow 0.458$

**Fig 1. This figure details the workflow for computing the new LiBlock measure and an example illustrating a use case of the workflow following the steps defined in algorithm 1.**

```
 3:    S₂ ← σ(s₂)                    ▷ word set sentence 2
 4:    D ← S₁ ∪ S₂                    ▷ construct the dictionary D
 5:    b₁ ← φ(S₁)                    ▷ construct the semantic binary vector b₁
 6:    b₂ ← φ(S₂)                    ▷ construct the semantic binary vector b₂
 7:    score_LiAd ← sim_LiAd(b₁, b₂)          ▷ compute LiAdapted
similarity
 8:    score_Bk ← sim_Bk(s₁, s₂)             ▷ compute Block Distance
similarity
 9:    score_LiBk ← sim_LiBk(score_LiAd, score_Bk)             ▷ compute LiBlock
similarity
10:    return score_LiBk
```

11: **end function**

$$sim_{LiBk} \quad : L_{\Sigma} \times L_{\Sigma} \to [0,1] \subset \mathbb{R}, \; L_{\Sigma} = \{\text{word sequences in alphabet } \Sigma\} \tag{1}$$

$$\text{(LiBlock similarity)}$$

$$sim_{LiBk}(s_1, s_2) \quad = \begin{cases} sim_{Bk}(s_1, s_2), & \text{if } sim_{LiAd}(\sigma(s_1), \sigma(s_2)) = 0 \\[2em] \dfrac{1}{2} sim_{Bk}(s_1, s_2) + \dfrac{1}{2} sim_{LiAd}(\sigma(s_1), \sigma(s_2)), & \text{otherwise} \end{cases} \tag{2}$$

$$sim_{Bk} \quad : L_{\Sigma} \times L_{\Sigma} \to [0,1] \subset \mathbb{R}, \qquad \text{(Block distance)}$$

$$sim_{Bk}(s_1, s_2) \quad = 1 - \frac{\sum\limits_{i=1}^{|D|} |fr(w_i, s_1) - fr(w_i, s_2)|}{\sum\limits_{i=1}^{|D|} fr(w_i, s_1 + s_2)}, \quad D = \sigma(s_1) \cup \sigma(s_2) \in \mathcal{P}(\Sigma) \tag{3}$$

$$sim_{LiAd} \quad : \mathcal{P}(D) \times \mathcal{P}(D) \to [0,1] \subset \mathbb{R}, \qquad \text{(Li's score adaptation)}$$

$$sim_{LiAd}(S_1, S_2) \quad = \frac{\varphi(S_1) \cdot \varphi(S_2)}{||\varphi(S_1)|| * ||\varphi(S_2)||}$$

$$\varphi \quad : \mathcal{P}(D) \to \{0,1\}^{|D|}, \qquad \text{(binary vector constructor)}$$

$$\varphi(S) \quad = (b_1, b_2, \ldots, b_{|D|}), \quad b_i = \begin{cases} 1, w_i \in D \\ 0, w_i \notin D \end{cases} \tag{4}$$

$$\sigma \quad : L_{\Sigma} \to \mathcal{P}(\Sigma), \qquad \text{(word set generator)}$$

$$\sigma(s) \quad = \{w \in \Sigma : \exists k \in [1, \text{length}(s)] \text{ such that } s_k = w\} \tag{5}$$

## The eight new variants of current ontology-based methods

The current family of ontology-based methods for biomedical sentence similarity proposed by Sogancioglu et al. [30] is based on the ontology-based semantic similarity between word and concepts within the sentences to be compared. Thus, this later family of methods defines a framework in which we can design new variants by exploring other word similarity measures. For this reason, we propose here the evaluation of a set of new ontology-based sentence simi-larity measures based on two different unexplored notions as follows: (1) the evaluation of

state-of-the-art word similarity measures from the general domain [48] not evaluated in the biomedical domain yet; and (2) the evaluation of several ontology-based word similarity measures based on a recent and very efficient shortest-path algorithm, called Ancestors-based Shortest-Path Length (AncSPL) [42], which is a fast approximation of the Dijkstra's algorithm [63] for taxonomies that is introduced with the first HESML version for the biomedical domain [42].

Thus, we propose here the evaluation based on the combination of WBSM and UBSM methods with the path-based word similarity methods as follows: WBSM-Rada (M7); WBSM-cosJ&C (M9); WBSM-coswJ&C (M10); WBSM-Cai (M11); UBSM-Rada (M12); UBSM-cosJ&C (M14); UBSM-coswJ&C (M15); and UBSM-Cai (M16). The detailed information about this later method is shown in Table 3.

### The new pre-trained word embedding model

Current sentence similarity methods based on the evaluation of pre-trained embedding models are mostly trained using PubMed Central (PMC) Open Access dataset (https://www.ncbi.nlm.nih.gov/labs/pmc/), or Medical Information Mart for Intensive Care (MIMIC-III) clinical notes [64]. However, as far as we know, there are no models in the literature trained on the full text of the articles in the PMC-BioC corpus [19]. Therefore, we propose evaluating a new FastText [58] word embedding model trained on the aforementioned BioC corpus. FastText overcomes one significant limitation of other methods, such as word2vec [65] and GloVe [66], which ignore the morphology of words by assigning a vector to each word in the vocabulary. For a more detailed review of the family of word embedding methods, we refer the authors to the recent reproducible survey by Lastra-Díaz et al. [48]. The configuration parameters for training this model are detailed in Table 4, and all the necessary information and resources for evaluating it are available in our reproducibility dataset [43], as detailed in Table 6.

### The reproducible experimental survey

This section introduces a detailed experimental setup to evaluate and compare all the sentence similarity methods for the biomedical domain proposed in our primary work [40], together with the new methods introduced herein. The main aims of our experiments are as follows: (1) the evaluation of most of known methods for biomedical sentence similarity on the three biomedical datasets shown in Table 1, and implemented on the same software platform; (2) the evaluation of a set of new sentence similarity methods adapted from their definitions for the general-language domain; (3) the evaluation of a new sentence method called LiBlock introduced in this work, eight variants of the current ontology-based methods from the literature based on the work of Sogancioglu et al. [30], and a new word embedding model based on FastText and trained on the full-text of articles in the PMC-BioC corpus [19]; (4) the setting out of the state of the art of the problem in a sound and reproducible way; (5) the replication and independent confirmation of previously reported methods and results; (6) a

**Table 1. Benchmarks on biomedical sentence similarity evaluated in this work.**

| Dataset | #pairs | Corresponding file (*.tsv) in HESML-STS distribution |
|---|---|---|
| BIOSSES [30] | 100 | BIOSSESNormalized.tsv |
| MedSTS [52] | 1,068 | CTRNormalized_averagedScore.tsv |
| CTR [53] | 170 | MedStsFullNormalized.tsv |

study on the impact of different pre-processing configurations on the performance of the sentence similarity methods; (7) a study on the impact of different Name Entity Recognition (NER) tools, such as MetaMap [34] and clinical Text Analysis and Knowledge Extraction System (cTAKES) [62], on the performance of the sentence similarity methods; (8) the evaluation for the first time of the CTR [53] dataset; (9) the identification of the main drawbacks and limitations of current methods; and finally, (10) a detailed statistical significance analysis of the results.

## Selection of methods

The criteria for the selection of the sentence similarity methods evaluated herein is as follows: (a) all the methods that have been evaluated in BIOSSES and MedSTS datasets; (b) a selection of methods that have not been evaluated in the biomedical domain yet; (c) a collection of new variants or adaptations of methods previously proposed for the general or biomedical domain, which are evaluated for the first time in this work, such as the WBSM-cosJ&C [30, 42, 46, 67], WBSM-coswJ&C [30, 42, 46, 67], WBSM-Cai [30, 42, 68], UBSM-cosJ&C [30, 42, 46, 67], UBSM-coswJ&C [30, 42, 46, 67], and UBSM-Cai [30, 42, 68] methods detailed in Tables 3 and 4; and (d) a new string-based method based on Li et al. [56] introduced in this work. For a more detailed description of the selection criteria of the methods, we refer the reader to our registered report protocol [40].

Tables 2 and 3 detail the configuration of the string-based measures and ontology-based measures that are evaluated here, respectively. Both WBSM and UBSM methods are evaluated in combination with the following word and concept similarity measures: Rada et al. [69], Jiang&Conrath [70], and three state-of-the-art unexplored measures, called cosJ&C [42, 46], coswJ&C [42, 46], and Cai et al. [42, 68]. The word similarity measure which reports the best results is used to evaluate the COM method [30, 69]. Table 4 details the sentence similarity methods based on the evaluation of pre-trained character, word, and Sentence Embedding (SE) models that are evaluated in this work. Finally, Table 5 details the pre-trained language models that are evaluated in our experiments.

**Table 2. Detailed setup for the string-based sentence similarity measures which are evaluated in this work.** All the string-based measures follow the implementation of Sogancioglu et al. [30], who use the Simmetrics library [71]. The LiBlock method proposed herein is an adaptation from Li et al. [56] combined with a string-based measure, as detailed in the previous section.

| ID | Method | Detailed setup of each method |
|---|---|---|
| M1 | Qgram [72] | $sim(a, b) = \frac{2 \times |q-grams(a) \cup q-grams(b)|}{|q-grams(a)| + |q-grams(b)|}$, being $a$ and $b$ sets of q words, and with q = 3. |
| M2 | Jaccard [73, 74] | $sim(a, b) = \frac{|a \cup b|}{|a \cap b|}$, being $a$ and $b$ sets of words of the first and second sentence respectively. |
| M3 | Block distance [57] | $sim(s_1, s_2) = 1 - \frac{\sum_{i=1}^{|D|} |fr(w_i, s_1) - fr(w_i, s_2)|}{\sum_{i=1}^{|D|} fr(w_i, s_1 + s_2)}$, as detailed in equation Eq 2. |
| M4 | LiBlock (this work) | LiBlock method (see Eq (1) annotated with CUI concepts and using cTAKES combined with the Block Distance [57] method using its best pre-processing configuration. |
| M5 | Levenshtein distance [75] | Measures the minimal cost number of insertions, deletions and replacements needed for transforming the first into the second sentence. Insert, delete and substitution cost set to 1. |
| M6 | Overlap coefficient [76] | $sim(a, b) = \frac{|a \cap b|}{|Min(|a|, |b|)|}$, being $a$ and $b$ sets of words of the first and second sentence respectively. |

**Table 3. Detailed setup for the ontology-based sentence similarity measures evaluated in this work.** The evaluation of the methods using Rada [69], coswJ&C [46], and Cai [68] word similarity measures use a reformulation of the original path-based measures based on the new Ancestors-based Shortest-Path Length (AncSPL) algorithm [42].

| ID | Sentence similarity method | Detailed setup of each method |
|---|---|---|
| M7 | WBSM-Rada [30, 42, 69] | WBSM [30] combined with Rada [69] measure using the AncSPL algorithm [42] |
| M8 | WBSM-J&C [30, 67, 70] | WBSM [30] combined with J&C [70]measure and Sanchez et al. [67] IC model |
| M9 | WBSM-cosJ&C [30, 42, 46] (this work) | WBSM [30] with cosJ&C [46] measure and Sanchez et al. [67] IC model using the AncSPL algorithm [42] |
| M10 | WBSM-coswJ&C [30, 42, 46, 67] (this work) | WBSM [30] with coswJ&C [46] measure and Sanchez et al. [67] IC model using the AncSPL algorithm [42] |
| M11 | WBSM-Cai [30, 42, 68] | WBSM [30] combined with Cai et al. [68] measure and Cai et al. [68] IC model using the AncSPL algorithm [42] |
| M12 | UBSM-Rada [30, 42, 69] | UBSM [30] with Rada et al. [69] measure using the AncSPL algorithm [42] |
| M13 | UBSM-J&C [30, 67, 70] | UBSM [30] combined with J&C [70] measure and Sanchez et al. [67] IC model |
| M14 | UBSM-cosJ&C [30, 47, 67] (this work) | UBSM [30] with cosJ&C [46] measure and Sanchez et al. [67] IC model using the AncSPL algorithm [42] |
| M15 | UBSM-coswJ&C [30, 42, 46, 67] (this work) | UBSM [30] with coswJ&C [46] measure and Sanchez et al. [67] IC model using the AncSPL algorithm [42] |
| M16 | UBSM-Cai [30, 42, 68] | UBSM [30] combined with Cai et al. [68] measure and Cai et al. [68] IC model using the AncSPL algorithm [42] |
| M17 | COM [30, 69] | $\lambda \cdot$ WBSM-Rada $+ (1 - \lambda) \cdot$ UBSM-Rada with $\lambda = 0.5$ |

**Table 4. Detailed setup for the sentence similarity methods based on pre-trained character, word (WE) and sentence (SE) embedding models evaluated herein.**

| ID | Sentence similarity method | Detailed setup of each method |
|---|---|---|
| M18 | Flair [77] | Contextual string embeddings trained on PubMed |
| M19 | Pyysalo et al. [78] | Skip-gram trained on PubMed + PMC |
| M20 | BioConceptVec [79] | Skip-gram WE model trained on PubMed using word2vec program |
| M21 | BioConceptVec [79] | CBOW WE model trained on PubMed using word2vec program |
| M22 | Newman-Griffis et al. [80] | Skip-gram WE model trained on PubMed using word2vec program |
| M23 | Newman-Griffis et al. [80] | CBOW WE model trained on PubMed using word2vec program |
| M24 | Newman-Griffis et al. [80] | GloVe WE model trained on PubMed |
| M25 | BioConceptVec$_{GloVe}$ [79] | GloVe We model trained on PubMed |
| M26 | BioWordVec$_{int}$ [81] | FastText [58] WE model trained on PubMed + MeSH |
| M27 | BioWordVec$_{ext}$ [81] | FastText [58] trained on PubMed + MeSH |
| M28 | BioNLP2016$_{win2}$ [82] | FastText [58] WE model based on skip-gram and trained on PubMed with training setup detailed in [82] |
| M29 | BioNLP2016$_{win30}$ [82] | FastText [58] WE model based on skip-gram and trained on PubMed with training setup detailed in [82] |
| M30 | BioConceptVec$_{fastText}$ [79] | FastText [58] WE model trained on PubMed |
| M31 | Universal Sentence Encoder (USE) [83] | USE SE pre-trained model of Cer et al. [83] |
| M32 | BioSentVec [33] | sent2vec [84] SE model trained on PubMed + MIMIC-III |
| M33 | FastText-Skipgram-BioC (this work) | FastText [58] WE model based on Skip-gram and trained on PMC-BioC corpus (05,09,2019) with the following setup: vector dim. = 200, learning rate = 0.05, sampling thres. = 1e-4, and negative examples = 10 |

**Table 5. Detailed setup for the sentence similarity methods based on pre-trained language models evaluated in this work.**

| ID | Sentence similarity method | Detailed setup of each method |
|---|---|---|
| M34 | BioBERT Base 1.0 [85] (+ PubMed) | BERT [86] trained on English Wikipedia + BooksCorpus + PubMed abstracts |
| M35 | BioBERT Base 1.0 [85] (+ PMC) | BERT [86] trained on English Wikipedia + BooksCorpus + PMC full-text articles |
| M36 | BioBERT Base 1.0 [85] (+ PubMed + PMC) | BERT [86] trained on English Wikipedia + BooksCorpus + PubMed abstracts + PMC full-text articles |
| M37 | BioBERT Base 1.1 [85] (+ PubMed) | BERT [86] trained on English Wikipedia + BooksCorpus + PubMed abstracts |
| M38 | BioBERT Large 1.1 [85] (+ PubMed) | BERT [86] trained on English Wikipedia + BooksCorpus + PubMed abstracts |
| M39 | NCBI-BlueBERT Base [32] PubMed | BERT [86] trained on PubMed abstracts |
| M40 | NCBI-BlueBERT Large [32] PubMed | BERT [86] trained on PubMed abstracts |
| M41 | NCBI-BlueBERT Base [32] PubMed + MIMIC-III | BERT [86] trained on PubMed abstracts + MIMIC-III |
| M42 | NCBI-BlueBERT Large [32] PubMed + MIMIC-III | BERT [86] trained on PubMed abstracts + MIMIC-III |
| M43 | SciBERT [87] | BERT [86] trained on PubMed abstracts |
| M44 | ClinicalBERT [88] | BERT [86] trained on PubMed abstracts |
| M45 | PubMedBERT [89] (abstracts) | BERT [86] trained on PubMed abstracts |
| M46 | PubMedBERT [89] (abstracts + full text) | BERT [86] trained on PubMed abstracts + full text |
| M47 | ouBioBERT-Base [90] (Uncased) | BERT [86] trained on PubMed abstracts |
| M48 | BioClinicalBERT [29] | BERT [86] trained on MIMIC-III |
| M49 | BioDischargesummaryBERT [29] | BERT [86] trained on MIMIC-III summaries |
| M50 | DischargesummaryBERT [29] | BERT [86] trained on MIMIC-III summaries |

## Pre-processing methods evaluated in this study

The pre-processing stage aims to ensure a fair comparison of the methods that are evaluated in a single end-to-end pipeline. To achieve this goal, the pre-processing stage normalizes and decomposes the sentences into a series of components that evaluate the same sequence of words applied to all the methods simultaneously. The selection criteria of the pre-processing components have been conditioned by the following constraints: (a) the pre-processing methods and tools used by state-of-the-art methods; and (b) the availability of resources and software tools. Fig 2 details all the possible combinations of pre-processing configurations that are

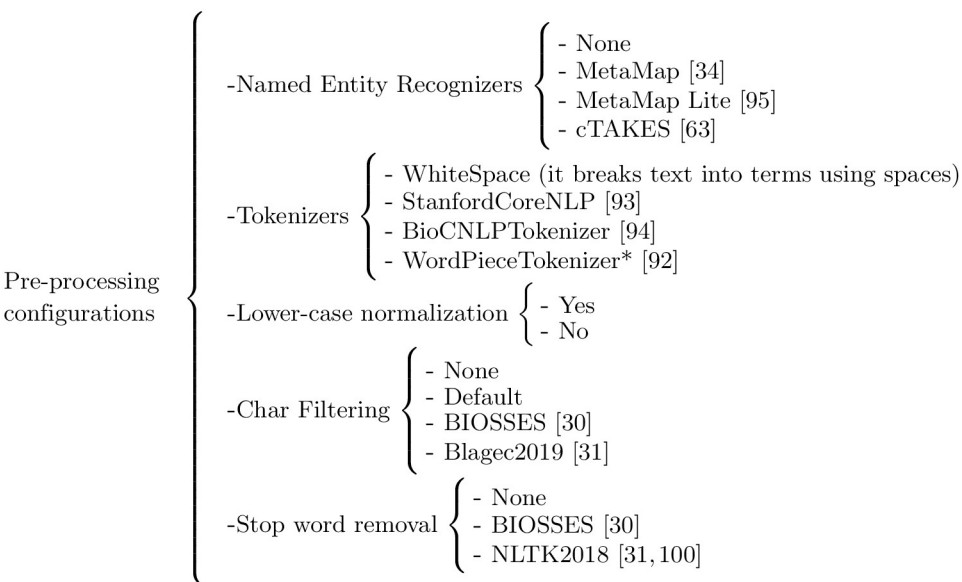

**Fig 2. Detail of the pre-processing configurations that are evaluated in this work.** (*) WordPieceTokenizer [91] is used only for BERT-based methods [30, 31, 34, 62, 91–94, 99].

evaluated in this work. String, word and sentence embedding, and ontology-based methods, are evaluated using all the available configurations except the WordPieceTokenizer [91], which is specific to BERT-based methods. Thus, BERT-based methods are evaluated using different char filtering, lower casing normalization, and stop word removal configurations. We use the Pearson and Spearman correlation metrics together with their harmonic score values to determine the impact of the different pre-processing configurations on the performance of the methods evaluated herein. However, we set the best overall performing pre-processing configuration using the harmonic average scores, as well as answering the remaining research questions.

Most methods receive as input the sequences of words making up the sentences to be compared. The process of splitting sentences into words can be carried out by tokenizers, such as the well-known general domain Stanford CoreNLP tokenizer [92], which is used by Blagec et al. [31], or the biomedical domain BioCNLPTokenizer [93]. On the other hand, the use of lexicons instead of tokenizers for sentence splitting would be inefficient because of the vast general and biomedical vocabulary. Besides, it would not be possible to provide a fair comparison of the methods because the pre-trained language models have no identical vocabularies.

The tokenized words that constitute the sentence, named tokens, are usually pre-processed by removing special characters and lower-casing, and removing the stop words. To analyze all the possible combinations of token pre-processing configurations from the literature, we replicate for each method those pre-processing configurations used by other authors, such as Blagec et al. [31] and Sogancioglu et al. [30], and we also evaluate all the pre-processing configurations that have not been evaluated yet. We also study the impact of the pre-processing configurations by not removing special characters and stop words from the tokens, nor normalizing them using lower-casing.

Ontology-based sentence similarity methods estimate the similarity of a sentence by exploiting the 'is-a' relationships between the concepts in an ontology. Therefore, the evaluation of any ontology-based method receives a set of concept-annotated pairs of sentences. The aim of the biomedical NER tools is to recognize automatically biomedical entities in pieces of raw text, such as diseases or drugs. We evaluate the impact of the three more broadly-used biomedical NER tools on the performance of the sentence similarity methods, as follows: (a) MetaMap [34], (b) cTAKES [62], and (c) MetaMap Lite [94]. MetaMap tool [34] is used by UBSM and COM methods [30] for recognizing Unified Medical Language System (UMLS) [95] concepts in the sentences, which is the standard compendium of biomedical vocabularies. Likewise, we use the default configuration of MetaMap restricted to the UMLS sources of SNOMED-CT and MeSH implemented by HESML V1R5 [42, 96], which is defined by the following features: (i) the use of all available semantic types; (ii) the MedPost Part-of-speech tagger [97]; and (iii) the MetaMap Word-Sense Disambiguation (WSD) module. We also evaluate cTAKES [63] because it has shown to be a robust and reliable tool to recognize biomedical entities [98]. Given the high computational cost of MetaMap in evaluating large text corpora, Demner-Fushman et al. [94] introduced a lighter MetaMap version, called Metamap Lite, which provides a real-time implementation of the basic MetaMap annotation capabilities without a large degradation of its performance.

Due to the large number of possible combinations of each pre-processing dimension, such as Named Entity Recognizers, tokenizers or char filtering methods, we have evaluated the pre-processing combinations of each dimension by defining a fixed pre-processing configuration for the rest of the dimensions, except for the string-based methods, whose performance is high enough to not cause a significant variation in the running time of the experiments.

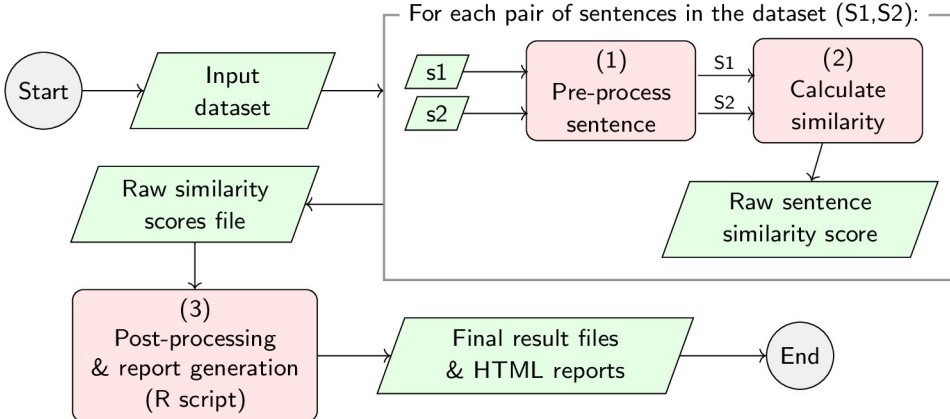

**Fig 3. Detailed workflow implemented by our experiments for pre-processing the input sentences, calculating the raw similarity scores, and post-processing the results obtained in the evaluation of the biomedical datasets.** This workflow generates a collection of raw and processed data files.

## Detailed workflow of our experiments

Fig 3 shows the workflow for running the experiments implemented in this work. Given an input dataset, such as BIOSSES [30], MedSTS [52], or CTR [53], the first step is to pre-process all the sentences, as shown in Fig 4. For each sentence pair ($s_1$, $s_2$) in the dataset, the pre-processing stage is divided into four stages as follows: (1.a) named entity recognition of UMLS [95] concepts, using different state-of-the-art NER tools, such as MetaMap [34] or cTAKES [62]; (1.b) tokenization of the sentences, using well-known tokenizers, such as the Stanford CoreNLP tokenizer [92], BioCNLPTokenizer [93], or WordPieceTokenizer [91] for BERT-based methods; (1.c) lower-case normalization; (1.d) character filtering, which allows the removal of punctuation marks or special characters; and finally, (1.e) the removal of stop-words, following different approximations evaluated by other authors like Blagec et al. [31] or Sogancioglu et al. [30]. Once each dataset is pre-processed in step 1 detailed in Fig 3), the aim of step 2 is to calculate the similarity score between each pair of sentences in the dataset to produce a raw output file containing all raw similarity scores, one score per sentence pair. Finally, a R-language script is used in step 3 to process the raw similarity files and produce the final human-readable tables reporting the Pearson and Spearman correlation values shown in Table 8, as well as the statistical significance of the results and any other supplementary data

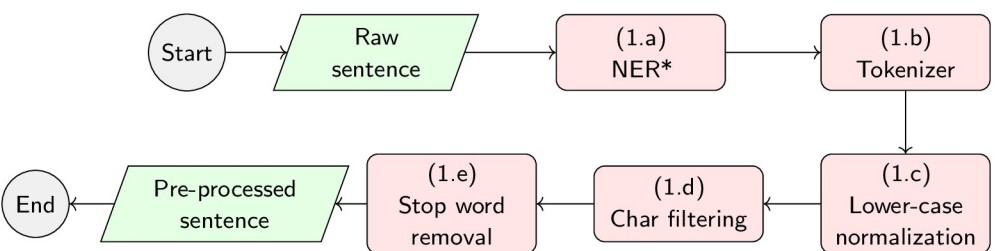

**Fig 4. Detailed sentence pre-processing workflow that are implemented in our experiments.** The pre-processing stage takes an input sentence and produces a pre-processed sentence as output. (*) The named entity recognizer are only evaluated in ontology-based methods.

table required by our study on the impact of the pre-processing and NER tools reported in appendices A and B respectively.

Finally, we also evaluate all the pre-processing combinations for each family of methods to study the impact of the pre-processing methods on the performance of the sentence similarity methods, with the only exception of the BERT-based methods. The pre-processing configurations of the BERT-based methods are only evaluated in combination with the WordPiece Tokenizer [91] because it is required by the current BERT implementations.

## Evaluation metrics

The evaluation metrics used to compare the performance of the methods analyzed are the following: (1) the Pearson correlation, denoted by $r$ in Eq (6); (2) the Spearman rank correlation, denoted by $\rho$ in equation (Eq (7)); (3) and the harmonic score, denoted by $h$ in equation (Eq (8)). The Pearson correlation evaluates the linear correlation between two random samples, whilst the Spearman rank correlation is rank-invariant and evaluates the monotonic relationship between two random samples, and the harmonic score allows comparing sentence similarity methods by using a single weighted score based on their performance in Pearson and Spearman correlation.

$$r \quad = \quad \frac{\sum_{i=1}^{n}(X_i - \bar{X})(Y_i - \bar{Y})}{\sqrt{\sum_{i=1}^{n}(X_i - \bar{X})^2}\sqrt{\sum_{i=1}^{n}(Y_i - \bar{Y})^2}} \tag{6}$$

$$\rho \quad = \quad 1 - \frac{6\sum_{i=1}^{n} d_i^2}{n(n^2 - 1)}, \qquad di = (x_i - y_i) \tag{7}$$

$$h \quad = \quad \frac{2r\rho}{r + \rho} \tag{8}$$

## Statistical significance of the results

We use the well-known t-Student test to carry-out a statistical significance analysis of the results of the evaluation of the methods in the tree biomedical datasets shown in Table 1. In order to compare the overall performance of the semantic measures that is evaluated in our experiments, we use the harmonic score average in all datasets. The statistical significance of the results is evaluated using the p-values resulting from the t-student test for the mean difference between the harmonic score values reported by each pair of semantic measures in all datasets. The p-values are computed using a one-sided t-student distribution on two paired random sample vectors made up of the harmonic ($h$) score values obtained in the evaluation of the three aforementioned datasets. Our null hypothesis, denoted by $H_0$, is that the difference in the average performance between each pair of compared sentence similarity methods is 0, whilst the alternative hypothesis, denoted by $H_1$, is that their average performance is different. For a 5% level of significance, it means that if the p-value is greater than or equal to 0.05, we must accept the null hypothesis. Otherwise, we can reject $H_0$ with an error probability of less than the p-value. In this latter case, we say that a first sentence similarity method obtains a statistically significantly higher value than the second one or that the former one significantly outperforms the second one.

**Uniform size datasets for our statistical significance analysis.** The scarcity of datasets for this problem and the notable size difference among datasets varying from 100 to 1,068 sentence pairs makes it impossible to study the statistical significance of the results with an

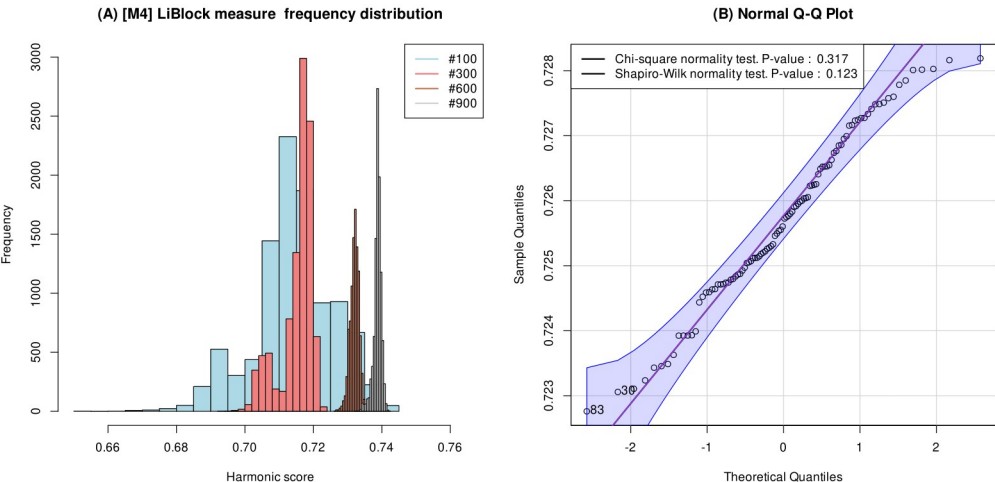

**Fig 5.** Figure (a) below shows the histogram plots for the harmonic score obtained by the Li-Block measure [M4] in evaluating the sentence similarity of 10,000 different equal-size subsets of sentence pairs extracted from the MedSTS dataset. Each histogram plot represents the frequency distribution of 10,000 samples of the harmonic score with subsets of sentence pairs with sizes: 100, 300, 600, and 900. Figure (b) shows the Q-Q plot normality test for the harmonic score obtained for a random subset with size 100, along with the p-values reported by the Shapiro-Wilk and Chi-square normality tests.

adequate sample size and to carry out a fair and unbiased comparison of the results. It is a known fact [48] that the statistical distribution of the Pearson and Spearman correlation values reported by any semantic similarity measure can significantly vary regarding the dataset size, which means that the statistical distribution of the harmonic score obtained for small subsets of a large dataset as MedSTS is not the same as that obtained for the whole dataset, as shown in Fig 5a. Fig 5a shows the histogram plots for the harmonic score obtained by the Li-Block measure [M4] in evaluating the sentence similarity of 10,000 different equal-size subsets of sentence pairs extracted from the MedSTS dataset for four different subset sizes: 100, 300, 600, and 900 sentence pairs. Fig 5a shows that the harmonic score follows a different normal distribution for each subset size, whose normality is subsequently confirmed by the Q-Q plot shown in Fig 5b and the Shilford-Wilk (p-value = 0.123) and Chi-square (p-value = 0.317) tests for the sample of harmonic score values for subsets with size 100. Thus, the correlation values derived from MedSTS (1,068 pairs) could bias our results and violate the underlying hypothesis of the t-Student test that requires that the data has the same normal distribution. This potential risk of degradation of our significance analysis increases by the fact that we only have 3 datasets of different sizes (100; 1,068; 170). For this reason, we have divided the MedSTS dataset into 10 parts, considered as independent datasets, to perform the study of the statistical significance of the results. Thus, we have artificially obtained 12 datasets of 100 to 200 pairs of sentences to build the vectors of harmonic score values used in the computation of the p-values. This set of datasets allows us to obtain the p-values to compare the statistical significance between the different measures, but does not affect the processed results from Table 8. All the necessary resources for obtaining both Table 8 and the table containing all the p-values reported in S1 Appendix are publicly available in the reproducibility dataset and the companion Lab Protocol article currently in preparation, as detailed in Table 6.

**Bonferroni correction for multiple hypothesis testing.** Our discussion introduces some conclusions derived from the evaluation of multiple pairwise hypothesis tests to elucidate the statistical significance of the outperformance of one baseline similarity measure among a family of methods. In these latter cases, we define a set of null hypotheses $\{H_1, \ldots, H_m\}$ setting that

**Table 6. Supplementary material and reproducibility resources of this work.**

| Material | Description |
|---|---|
| Reproducibility dataset [43] | All raw input and output data files, pre-trained model files, and a long-term reproducibility image based on Docker, which is publicly available on the Spanish Dataverse Network (https://doi.org/10.21950/EPNXTR) |
| Reproducibility protocol [44] | Raw step-by-step instructions to download the required resources and reproduce the experiments evaluated in this work |
| Lab Protocol article [44] (under preparation) | Data and methods article introducing a very detailed description of our experiments, datasets, and reproducibility protocol to allow the independent replication of our experiments and results |
| HESML-STS software library (integrated into HESML V2R1) | Release of the new HESML-STS library. This library is based on the previous HESML V1R5 version [41, 42] published in Github (https://github.com/jjlastra/HESML) and the Spanish Dataverse Network [43] under a CC By-NC-SA-4.0 license. |
| HESML V2R1 software release [105] | Release of the new HESML V2R1 version. This new release is based on the previous HESML V1R5 version [42], including the new HESML-STS software package that has been developed for this study, after managing all the licensing restrictions of the NER tools. |
| HESML-STS software paper [104] (under preparation) | Software article introducing our sentence similarity library, called HESML-STS, together with some benchmarks under preparation. |

the pairwise mean difference between the harmonic score obtained by one baseline measure and the remaining methods in the same family is 0. To reduce the family-wise type I error (false positives) derived from our multiple comparisons [100], we define a Bonferroni correction to evaluate the statistical significance of multiple hypothesis tests involved in those conclusions in which one baseline sentence similarity measure is compared with a family of methods. For each single conclusion comparing one baseline measure with other methods, we define a corrected null-hypothesis rejection threshold $\alpha_c$ defined as $\alpha_c = \alpha/m$, where $\alpha$ is equal to 0.05 for a 5% level of significance and $m$ is the number of pairwise comparisons (uncorrected p-values). Thus, the null-hypothesis is only rejected if the p-values are lower than $\alpha_c$ when multiple pairwise hypotheses are tested.

## Statistical performance analysis of the best methods

In order to answer the RQ5 research question, we study how well the sentence similarity methods estimate the degree of semantic similarity between two sentences by analyzing the deviation of their estimated values with respect to the human similarity scores. We want to analyze why the methods are doing well or badly on specific sentence pairs to provide an explanation for this behaviour, as well as identifying the main drawbacks and limitations of the current state-of-the-art methods. To carry out this performance analysis, we analyze the statistics of the similarity error function $E_{sim}$ of the methods defined in Eq 9. We only use some sentences extracted from the BIOSSES dataset for this analysis because this dataset has no licensing restrictions on its use, which allows us to reproduce their sentences here, unlike MedSTS. We could have also used CTR because it has no licensing restrictions; however, CTR has not been previously used in this sentence similarity task.

$$
\begin{aligned}
E_{sim} \quad &: L_\Sigma \times L_\Sigma \to [0,1] \subset \mathbb{R} \\
E_{sim}(s_1, s_2) \quad &= sim(s_1, s_2) - humanSim(s_1, s_2)
\end{aligned}
\tag{9}
$$

Our methodology to conduct the performance analysis is detailed below:

1. Selection of the best-performing method from each family of methods.

2. Estimation of the Probability Density Function (PDF) of the $E_{sim}$ function for the evaluation of the selected best-performing methods in each dataset by calling the "*density*" function provided by the R statistical package.

3. Selection of the sentences based on their similarity error in the BIOSSES dataset:

   3.1 the sentences with the lowest and highest absolute similarity error $|E_{sim}|$ for each method are extracted.

   3.2 each sentence selected in the step above is pre-processed using the best pre-processing configuration for each method.

   3.3 the resulting pre-processed sentences and the statistical information of the similarity scores are analyzed in the *Discussion* section.

## Software implementation

We have developed a new sentence measures library for the biomedical domain called HESML-STS, which is based on HESML V1R5 [41, 42], as detailed in Table 6. All our experiments are generated by running the *HESMLSTSclient* and *HESMLSTSImpactpre-processingclient* programs, which generates a raw output file in comma-separated file format (*.csv) for each dataset detailed in Table 1. The raw output files contain the raw similarity values returned by each sentence similarity method in the evaluation of the degree of similarity between sentences. The final results for the Pearson and Spearman correlation, and the harmonic values detailed in Table 8 are automatically generated by running an R-language script file on the collection of raw similarity files, which also generates all the tables reported in appendices A and B provided as supplementary material. All tables are written both in LaTeX and comma-separated file format (*.csv) formats. For a more detailed description of the protocol for running our experiments, we refer the reader to the protocol [44] detailed in S3 Appendix.

We implemented a parser for loading pre-trained embedding models based on FastText [58] and other word embedding models [78–82], which are efficiently evaluated as sentence similarity measures in HESML by implementing the averaging Simple Word EMbedding (SWEM) approach introduced by Shen et al. [101]. However, the software replication required to evaluate sentence embedding and BERT-based language models is extremely complex and out of the scope of this work. For this reason, these models are evaluated using the original software artifacts used to generate the aforementioned pre-trained models. Thus, we implemented a collection of Python wrappers for evaluating the available models by using the provided software artifacts as follows: (1) Sent2vec-based models [33] are evaluated using the Sent2vec library [84]; (2) Flair models [77] are evaluated using the flairNLP framework [77]; and USE models [83] are evaluated using the open source platform TensorFlow [102]. All BERT-based pre-trained models are evaluated using the open source bert-as-a-service library [103].

## Reproducing our benchmarks

For the sake of reproducibility, we introduce a detailed reproducibility protocol on protocols. io [44] that is based on a reproducibility dataset [43] containing all the software and data necessary to allow the exact replication of all our experiments and results. Our reproducibility protocol is mainly based on a Docker-based image that includes a pre-installation of all the necessary software and the Java source code and binary files of our benchmark program, which is provided as supplementary material in our reproducibility dataset [43] and

DockerHub (https://hub.docker.com/repository/docker/alicialara/hesml_v2r1). Our source code files are tagged on Github with a permanent tag named "Release_HESML_V2R1" (https://github.com/jjlastra/HESML/releases/tag/Release_HESML_V2R1).

In addition, we plan to submit a Lab Protocol article under preparation [44] (https://collections.plos.org/collection/lab-protocols), which will provide a detailed description of the publicly available reproducibility dataset [43] and a very detailed reproducibility protocol [44] to allow the exact replication of all our methods, experiments, and results. We also plan to submit another article [104], currently in preparation, to introduce the new HESML-STS software library integrated into the latest HESML V2R1 version [105], together with a set of reproducible benchmarks on semantic measures libraries for biomedical sentence similarity. However, our reproducibility dataset allows the full and exact replication of all our experiments by completing the licensing requirements of the UMLS databases and the aforementioned NER tools for the National Library of Medicine (NLM) of the United States (https://www.nlm.nih.gov/databases/umls.html#license_request).

Table 6 details all the reproducibility resources provided as supplementary material with this work. Our benchmarks are implemented using Java 8, Python 3 and R programming languages, and thus, they can be reproduced in any Java-compliant or Docker-compliant platforms, such as Windows, MacOS, or any Linux-based system.

## Results obtained

Table 7 shows the selected pre-processing configuration of each method for obtaining their best-performing results, whilst Table 8 shows the results obtained in the evaluation of all methods in the three biomedical datasets evaluated herein by using their best pre-processing configurations. Table 9 shows the comparison of results for the highest (best) and lowest (worst) average harmonic score values for the best-performing method of each family shown in blue in Table 8, which are defined by the method obtaining the highest average harmonic score. Furthermore, Table 10 shows the results obtained in our study on the impact of NER tools on the performance of the sentence similarity methods in the evaluation of the MedSTS dataset [52]. Table 11 shows the harmonic and average harmonic scores obtained in the evaluation of the three biomedical datasets, as well as the resulting p-values comparing the NER tools for each ontology-based method. Table 12 shows the results obtained in the evaluation of the LiBlock method in the three biomedical datasets by using its best pre-processing configuration, and annotating the sentences with all the NER tools combinations. In addition, the aforementioned table details the resulting p-values comparing the best-performing LiBlock-NER combination with the other NER tools. Tables 13–16 show the raw input sentence pairs and their corresponding pre-processed versions in which the best-performing methods obtain the lowest and highest similarity error ($E_{sim}$) in the BIOSSES dataset [30]. Table 17 details the statistical information for the best-performing methods of each family in the evaluation of the three biomedical datasets evaluated in this study. Finally, Fig 6 shows the Probability Density Function (PDF) of the similarity error obtained by the best-performing methods of each family in the evaluation of the BIOSSES, MedSTS, and CTR datasets respectively.

S1 Appendix shows the p-values resulting from comparing all the methods using their best pre-processing configuration as detailed in Table 8, which allows us to study the statistical significance of the results, as detailed in the Discussion section. In addition, appendix B shows the experimental results regarding the impact of pre-processing configurations in all the methods evaluated here; the best configuration has been used to determine the final scores for each method. Finally, S3 Appendix details the protocol for reproducing all the experiments evaluated in this paper, and is also published on protocols.io [44].

**Table 7. Best-performing pre-processing configurations used to evaluate the methods compared in this work as reported in Table 8, derived from our cross-evaluation of each method with the pre-processing configurations shown in Fig 2 (see S2 Appendix).** (*) COM (M17) uses the best configuration of the WBSM-Rada (M7) and UBSM-Rada (M12) methods for computing the similarity scores.

| ID | Sentence similarity method | NER | Tokenizer | Lower-case | Char filtering | Stop words removal |
|---|---|---|---|---|---|---|
| M1 | Qgram | None | WhiteSpace | yes | BIOSSES | NLTK2018 |
| M2 | Jaccard | None | WhiteSpace | yes | BIOSSES | NLTK2018 |
| M3 | Block distance | None | WhiteSpace | yes | BIOSSES | NLTK2018 |
| M4 | LiBlock (this work) | cTAKES | CoreNLP | yes | Default | NLTK2018 |
| M5 | Levenshtein distance | None | WhiteSpace | no | None | BIOSSES |
| M6 | Overlap coefficient | None | CoreNLP | yes | Default | NLTK2018 |
| M7 | WBSM-Rada | Exact matching | CoreNLP | yes | BIOSSES | NLTK2018 |
| M8 | WBSM-J&C | Exact matching | CoreNLP | yes | BIOSSES | None |
| M9 | WBSM-cosJ&C (this work) | Exact matching | CoreNLP | yes | BIOSSES | None |
| M10 | WBSM-coswJ&C (this work) | Exact matching | CoreNLP | yes | BIOSSES | NLTK2018 |
| M11 | WBSM-Cai | Exact matching | CoreNLP | yes | BIOSSES | None |
| M12 | UBSM-Rada | cTAKES | CoreNLP | yes | BIOSSES | NLTK2018 |
| M13 | UBSM-J&C | MetamapLite | CoreNLP | yes | BIOSSES | NLTK2018 |
| M14 | UBSM-cosJ&C (this work) | MetamapLite | CoreNLP | yes | BIOSSES | NLTK2018 |
| M15 | UBSM-coswJ&C (this work) | cTAKES | CoreNLP | yes | BIOSSES | NLTK2018 |
| M16 | UBSM-Cai | MetamapLite | CoreNLP | yes | BIOSSES | NLTK2018 |
| M17 | COM (*) | - | - | - | - | - |
| M18 | Flair | None | WhiteSpace | no | BIOSSES | None |
| M19 | Pyysalo et al. | None | CoreNLP | yes | Default | BIOSSES |
| M20 | BioConceptVec$_{word2vec\_sg}$ | None | CoreNLP | yes | Default | BIOSSES |
| M21 | BioConceptVec$_{word2vec\_cbow}$ | None | CoreNLP | yes | Default | BIOSSES |
| M22 | Newman-Griffis$_{word2vec\_sgns}$ | None | CoreNLP | yes | Default | NLTK2018 |
| M23 | Newman-Griffis$_{word2vec\_cbow}$ | None | CoreNLP | yes | Default | NLTK2018 |
| M24 | Newman-Griffis$_{glove}$ | None | CoreNLP | yes | Default | NLTK2018 |
| M25 | BioConceptVec$_{glove}$ | None | CoreNLP | yes | Default | BIOSSES |
| M26 | BioWordVec$_{int}$ | None | CoreNLP | yes | BIOSSES | None |
| M27 | BioWordVec$_{ext}$ | None | CoreNLP | yes | BIOSSES | None |
| M28 | BioNLP2016$_{win2}$ | None | CoreNLP | no | Default | NLTK2018 |
| M29 | BioNLP2016$_{win30}$ | None | CoreNLP | no | Default | NLTK2018 |
| M30 | BioConceptVec$_{fastText}$ | None | CoreNLP | yes | Default | BIOSSES |
| M31 | USE | None | CoreNLP | no | Default | None |
| M32 | BioSentVec (PubMed+MIMIC-III) | None | CoreNLP | yes | BIOSSES | BIOSSES |
| M33 | FastText-SkGr-BioC (this work) | None | CoreNLP | yes | Default | None |
| M34 | BioBERT Base 1.0 (+ PubMed) | None | WordPiece | yes | BIOSSES | None |
| M35 | BioBERT Base 1.0 (+ PMC) | None | WordPiece | yes | BIOSSES | None |
| M36 | BioBERT Base 1.0 (PubMed+PMC) | None | WordPiece | yes | BIOSSES | None |
| M37 | BioBERT Base 1.1 (+ PubMed) | None | WordPiece | no | Blagec2019 | NLTK2018 |
| M38 | BioBERT Large 1.1 (+ PubMed) | None | WordPiece | no | Blagec2019 | NLTK2018 |
| M39 | NCBI-BlueBERT Base PubMed | None | WordPiece | yes | Blagec2019 | None |
| M40 | NCBI-BlueBERT Large PubMed | None | WordPiece | yes | BIOSSES | None |
| M41 | NCBI-BlueBERT Base PubMed + MIMIC-III | None | WordPiece | yes | BIOSSES | BIOSSES |
| M42 | NCBI-BlueBERT Large PubMed + MIMIC-III | None | WordPiece | yes | BIOSSES | None |
| M43 | SciBERT | None | WordPiece | yes | BIOSSES | NLTK2018 |
| M44 | ClinicalBERT | None | WordPiece | no | Blagec2019 | BIOSSES |
| M45 | PubMedBERT (abstracts) | None | WordPiece | yes | Default | NLTK2018 |
| M46 | PubMedBERT (abstracts+full text) | None | WordPiece | yes | Default | NLTK2018 |

(*Continued*)

**Table 7.** (Continued)

| ID | Sentence similarity method | NER | Tokenizer | Lower-case | Char filtering | Stop words removal |
|---|---|---|---|---|---|---|
| M47 | ouBioBERT-Base, Uncased | None | WordPiece | yes | Default | None |
| M48 | BioClinicalBERT | None | WordPiece | yes | Blagec2019 | BIOSSES |
| M49 | BioDischargesummaryBERT | None | WordPiece | no | Blagec2019 | NLTK2018 |
| M50 | DischargesummaryBERT | None | WordPiece | no | Blagec2019 | NLTK2018 |

## Discussion

### Comparison of string-based methods

*LiBlock (M4) obtains the highest average harmonic score among the family of string-based methods and significantly outperforms all of them.* This conclusion can be drawn by looking at the average column in Table 8 for this group of methods and checking the p-values reported in Table A.1 in S1 Appendix. Table A.1 in S1 Appendix shows that LiBlock obtains p-values lower than $\alpha_c$ = 0.05/5 (0,01) when it is compared with all the string-based methods, such as Block Distance (p-value = 0.000), Jaccard (p-value = 0.000), QGram (p-value = 0.000), Overlap Coefficient (p-value = 0.000), and Levenshtein (p-value = 0.000).

*LiBlock (M4) obtains the highest Pearson correlation value in the BIOSSES and MedSTS datasets among the family of string-based methods, whilst Block Distance (M3) obtains the highest Pearson correlation in the CTR dataset.* This conclusion can be drawn by looking at the results for the first group of methods detailed in Table 8.

*LiBlock (M4) obtains the highest Spearman correlation value in all datasets among the family of string-based methods.* This conclusion can be drawn by looking at the results for the first group of methods detailed in Table 8.

*LiBlock (M4) obtains the highest harmonic score in all datasets among the family of string-based methods.* This conclusion can be drawn by looking at the results for the first group of methods detailed in Table 8.

### Comparison of Ontology-based methods

*COM (M17) obtains the highest average harmonic score among the family of ontology-based methods and significantly outperforms all of them, with the sole exception of WBSM-Rada (M7).* This conclusion can be drawn by looking at the average column in Table 8 for the second group of methods and checking the p-values shown in Table A.1 in S1 Appendix. Table A.1 in S1 Appendix shows that COM obtains a p-value lower than $\alpha_c$ = 0.05/10 (0,005) when it is compared with all ontology-based methods, with the only exception of WBSM-Rada (M7) (p-value = 0.088).

*COM (M17) obtains the highest Pearson correlation value in the BIOSSES and CTR datasets among the family of ontology-based methods, whilst the WBSM-Rada (M7) methods obtain the highest Pearson correlation value in the MedSTS dataset.* This conclusion can be drawn by looking at the second group of methods in 8.

*COM (M17) obtains the highest Spearman correlation values in the BIOSSES dataset among the family of ontology-based methods, whilst WBSM-Rada (M7) and UBSM-Rada (M12) do so in the MedSTS and CTR datasets, respectively.* This conclusion can be drawn by looking at the second group of methods in 8.

*COM (M17) obtains the highest harmonic score in the BIOSSES and CTR datasets among the family of ontology-based methods, whilst WBSM-Rada (M7) does so in the MedSTS dataset.* This conclusion can be drawn by looking at the second group of methods detailed in Table 8.

**Table 8. Pearson (r), Spearman (ρ), harmonic (h), and harmonic average (AVG) scores obtained by each sentence similarity method evaluated herein in the three biomedical sentence similarity benchmarks arranged by families.** All reported values were obtained using the best pre-processing configurations detailed in Table 7. The results in bold show the best scores whilst results in blue show the best average harmonic score for each family.

| ID | Sentence similarity methods | BIOSSES [30] | | | MedSTS$_{full}$ [52] | | | CTR [53] | | | AVG |
|---|---|---|---|---|---|---|---|---|---|---|---|
| | | r | ρ | h | r | ρ | h | r | ρ | h | h |
| M1 | Qgram | 0.752 | 0.773 | 0.763 | 0.701 | 0.674 | 0.687 | 0.763 | 0.766 | 0.764 | 0.738 |
| M2 | Jaccard | 0.782 | 0.815 | 0.798 | 0.706 | 0.680 | 0.693 | 0.759 | 0.797 | 0.777 | 0.756 |
| M3 | Block distance | 0.798 | 0.818 | 0.808 | 0.731 | 0.683 | 0.706 | 0.797 | 0.801 | 0.799 | 0.771 |
| M4 | LiBlock (this work) | 0.820 | **0.828** | **0.824** | 0.769 | **0.710** | 0.739 | 0.793 | 0.808 | 0.800 | **0.788** |
| M5 | Levenshtein distance | 0.529 | 0.536 | 0.533 | 0.610 | 0.634 | 0.622 | 0.498 | 0.536 | 0.516 | 0.557 |
| M6 | Overlap coefficient | 0.782 | 0.795 | 0.788 | 0.696 | 0.564 | 0.623 | 0.781 | 0.793 | 0.787 | 0.733 |
| M7 | WBSM-Rada | 0.772 | 0.791 | 0.782 | **0.774** | 0.709 | **0.740** | 0.785 | 0.765 | 0.775 | 0.766 |
| M8 | WBSM-J&C | 0.483 | 0.549 | 0.514 | 0.647 | 0.614 | 0.630 | 0.536 | 0.516 | 0.526 | 0.557 |
| M9 | WBSM-cosJ&C (this work) | 0.483 | 0.549 | 0.514 | 0.647 | 0.614 | 0.630 | 0.536 | 0.516 | 0.526 | 0.557 |
| M10 | WBSM-coswJ&C (this work) | 0.571 | 0.566 | 0.568 | 0.705 | 0.651 | 0.677 | 0.637 | 0.590 | 0.613 | 0.619 |
| M11 | WBSM-Cai | 0.458 | 0.542 | 0.497 | 0.629 | 0.601 | 0.615 | 0.492 | 0.459 | 0.475 | 0.529 |
| M12 | UBSM-Rada | 0.792 | 0.809 | 0.800 | 0.763 | 0.700 | 0.730 | 0.776 | 0.794 | 0.785 | 0.772 |
| M13 | UBSM-J&C | 0.529 | 0.573 | 0.550 | 0.683 | 0.621 | 0.650 | 0.620 | 0.585 | 0.602 | 0.601 |
| M14 | UBSM-cosJ&C (this work) | 0.615 | 0.648 | 0.631 | 0.699 | 0.638 | 0.667 | 0.709 | 0.646 | 0.676 | 0.658 |
| M15 | UBSM-coswJ&C (this work) | 0.730 | 0.769 | 0.749 | 0.697 | 0.625 | 0.659 | 0.713 | 0.673 | 0.693 | 0.700 |
| M16 | UBSM-Cai | 0.545 | 0.579 | 0.562 | 0.686 | 0.628 | 0.656 | 0.642 | 0.576 | 0.607 | 0.608 |
| M17 | COM | 0.793 | 0.809 | 0.801 | 0.773 | 0.708 | 0.739 | 0.789 | 0.783 | 0.786 | 0.776 |
| M18 | Flair | 0.628 | 0.625 | 0.626 | -0.014 | -0.035 | -0.020 | 0.652 | 0.719 | 0.684 | 0.430 |
| M19 | Pyysalo et al. [78] | 0.713 | 0.706 | 0.709 | 0.754 | 0.641 | 0.693 | 0.744 | 0.803 | 0.773 | 0.725 |
| M20 | BioConceptVec$_{word2vec\_sg}$ | 0.742 | 0.743 | 0.742 | 0.751 | 0.652 | 0.698 | 0.738 | 0.800 | 0.768 | 0.736 |
| M21 | BioConceptVec$_{word2vec\_cbow}$ | 0.670 | 0.655 | 0.662 | 0.746 | 0.650 | 0.695 | 0.659 | 0.714 | 0.685 | 0.681 |
| M22 | Newman-Griffis$_{word2vec\_sgns}$ | 0.771 | 0.763 | 0.767 | 0.764 | 0.641 | 0.697 | **0.799** | **0.835** | **0.817** | 0.760 |
| M23 | Newman-Griffis$_{word2vec\_cbow}$ | 0.675 | 0.686 | 0.681 | 0.746 | 0.647 | 0.693 | 0.697 | 0.768 | 0.731 | 0.701 |
| M24 | Newman-Griffis$_{glove}$ | 0.671 | 0.678 | 0.674 | 0.740 | 0.643 | 0.688 | 0.732 | 0.729 | 0.731 | 0.698 |
| M25 | BioConceptVec$_{glove}$ | 0.547 | 0.585 | 0.565 | 0.720 | 0.648 | 0.682 | 0.624 | 0.694 | 0.657 | 0.635 |
| M26 | BioWordVec$_{int}$ | **0.831** | 0.806 | 0.818 | 0.766 | 0.686 | 0.724 | 0.757 | 0.735 | 0.746 | 0.763 |
| M27 | BioWordVec$_{ext}$ | 0.752 | 0.725 | 0.738 | 0.756 | 0.673 | 0.712 | 0.736 | 0.729 | 0.732 | 0.727 |
| M28 | BioNLP2016$_{win2}$ | 0.697 | 0.693 | 0.695 | 0.699 | 0.594 | 0.642 | 0.691 | 0.759 | 0.724 | 0.687 |
| M29 | BioNLP2016$_{win30}$ | 0.745 | 0.751 | 0.748 | 0.714 | 0.609 | 0.657 | 0.742 | 0.810 | 0.774 | 0.727 |
| M30 | BioConceptVec$_{fastText}$ | 0.091 | 0.262 | 0.135 | 0.416 | 0.456 | 0.435 | 0.178 | 0.264 | 0.212 | 0.261 |
| M31 | USE | 0.666 | 0.669 | 0.668 | 0.679 | 0.606 | 0.640 | 0.663 | 0.684 | 0.674 | 0.660 |
| M32 | BioSentVec | 0.797 | 0.767 | 0.782 | 0.763 | 0.638 | 0.695 | 0.791 | 0.821 | 0.806 | 0.761 |
| M33 | FastText-SkGr-BioC (this work) | 0.814 | 0.777 | 0.795 | 0.758 | 0.660 | 0.706 | 0.761 | 0.760 | 0.760 | 0.754 |
| M34 | BioBERT Base 1.0 (+ PubMed) | 0.569 | 0.567 | 0.568 | 0.662 | 0.576 | 0.616 | 0.616 | 0.642 | 0.629 | 0.604 |
| M35 | BioBERT Base 1.0 (+ PMC) | 0.664 | 0.663 | 0.664 | 0.674 | 0.581 | 0.624 | 0.601 | 0.647 | 0.623 | 0.637 |
| M36 | BioBERT Base 1.0$_{(PubMed+ PMC)}$ | 0.616 | 0.609 | 0.612 | 0.647 | 0.561 | 0.601 | 0.638 | 0.663 | 0.650 | 0.621 |
| M37 | BioBERT Base 1.1 (+ PubMed) | 0.668 | 0.647 | 0.657 | 0.712 | 0.616 | 0.661 | 0.643 | 0.663 | 0.653 | 0.657 |
| M38 | BioBERT Large 1.1 (+ PubMed) | 0.557 | 0.546 | 0.551 | 0.695 | 0.622 | 0.657 | 0.579 | 0.650 | 0.612 | 0.607 |
| M39 | NCBI-BlueBERT Base PubMed | 0.682 | 0.668 | 0.675 | 0.679 | 0.565 | 0.617 | 0.668 | 0.719 | 0.693 | 0.662 |
| M40 | NCBI-BlueBERT Large PubMed | 0.688 | 0.712 | 0.700 | 0.636 | 0.588 | 0.611 | 0.609 | 0.674 | 0.640 | 0.650 |
| M41 | NCBI-BlueBERT Base PubMed + MIMIC-III | 0.537 | 0.536 | 0.536 | 0.733 | 0.624 | 0.674 | 0.548 | 0.553 | 0.550 | 0.587 |
| M42 | NCBI-BlueBERT Large PubMed + MIMIC-III | 0.560 | 0.578 | 0.569 | 0.675 | 0.628 | 0.651 | 0.487 | 0.504 | 0.496 | 0.572 |
| M43 | SciBERT | 0.653 | 0.616 | 0.634 | 0.727 | 0.643 | 0.683 | 0.604 | 0.682 | 0.641 | 0.652 |
| M44 | ClinicalBERT | 0.415 | 0.483 | 0.447 | 0.652 | 0.566 | 0.606 | 0.470 | 0.500 | 0.485 | 0.512 |
| M45 | PubMedBERT (abstracts) | 0.502 | 0.524 | 0.513 | 0.626 | 0.531 | 0.575 | 0.479 | 0.645 | 0.550 | 0.546 |

*(Continued)*

**Table 8.** (Continued)

| ID | Sentence similarity methods | BIOSSES [30] | | | MedSTS$_{full}$ [52] | | | CTR [53] | | | AVG |
|---|---|---|---|---|---|---|---|---|---|---|---|
| | | r | ρ | h | r | ρ | h | r | ρ | h | h |
| M46 | PubMedBERT (abstracts+full text) | 0.659 | 0.651 | 0.655 | 0.712 | 0.590 | 0.645 | 0.596 | 0.675 | 0.633 | 0.644 |
| M47 | ouBioBERT-Base, Uncased | 0.687 | 0.729 | 0.707 | 0.707 | 0.583 | 0.639 | 0.670 | 0.694 | 0.682 | 0.676 |
| M48 | BioClinicalBERT | 0.416 | 0.447 | 0.431 | 0.646 | 0.562 | 0.601 | 0.472 | 0.478 | 0.475 | 0.502 |
| M49 | BioDischargesummaryBERT | 0.376 | 0.397 | 0.387 | 0.637 | 0.565 | 0.599 | 0.385 | 0.465 | 0.421 | 0.469 |
| M50 | DischargesummaryBERT | 0.395 | 0.465 | 0.427 | 0.655 | 0.567 | 0.608 | 0.376 | 0.407 | 0.391 | 0.475 |

**Table 9. Comparison of results for the "best" and the "worst" pre-processing configurations for the best-performing methods of each family in Table 8.** The last column shows the t-Student p-values comparing the best and worst configurations.

| ID | Methods | Pre-processing configuration | BIOSSES | | | MedSTS$_{full}$ | | | CTR | | | AVG | p-val |
|---|---|---|---|---|---|---|---|---|---|---|---|---|---|
| | | | r | ρ | h | r | ρ | h | r | ρ | h | h | |
| M4 | LiBlock (worst) | TOK-Whitespace LC-No SW-NLTK2018 CF-None | 0.779 | 0.793 | 0.786 | 0.736 | 0.676 | 0.704 | 0.765 | 0.717 | 0.741 | 0.744 | |
| | | | | | | | | | | | | | 0.000 |
| M4 | LiBlock (best) | TOK-CoreNLP LC-Yes SW-NLTK2018 CF-Default | 0.820 | 0.828 | 0.824 | 0.769 | 0.710 | 0.739 | 0.793 | 0.808 | 0.800 | 0.788 | |
| M17 | COM (worst) | —WBSM-Rada - UBSM-Rada (worst): TOK-Whitespace LC-Yes SW-None CF-None | 0.610 | 0.635 | 0.622 | 0.681 | 0.648 | 0.664 | 0.656 | 0.662 | 0.659 | 0.648 | |
| | | | | | | | | | | | | | 0.000 |
| M17 | COM (best) | —WBSM-Rada - UBSM-Rada (best): TOK-CoreNLP LC-Yes SW-NLTK2018 CF-BIOSSES | 0.793 | 0.809 | 0.801 | 0.773 | 0.708 | 0.739 | 0.789 | 0.783 | 0.786 | 0.776 | |
| M26 | BioWordVec$_{int}$ (worst) | TOK-Whitespace LC-No SW-None CF-None Pooling-Sum | 0.436 | 0.497 | 0.465 | 0.532 | 0.619 | 0.572 | 0.529 | 0.674 | 0.593 | 0.543 | |
| | | | | | | | | | | | | | 0.000 |
| M26 | BioWordVec$_{int}$ (best) | TOK-CoreNLP LC-Yes SW-None CF-BIOSSES Pooling-Min | 0.831 | 0.809 | 0.820 | 0.764 | 0.682 | 0.721 | 0.761 | 0.736 | 0.748 | 0.763 | |
| M47 | OuBioBert (worst) | TOK- WordPiece LC-Yes SW-BIOSSES CF-Default | 0.608 | 0.627 | 0.617 | 0.730 | 0.622 | 0.672 | 0.669 | 0.696 | 0.682 | 0.657 | |
| | | | | | | | | | | | | | 0.000 |
| M47 | OuBioBert (best) | TOK-WordPiece LC-Yes SW-None CF-Default | 0.687 | 0.729 | 0.707 | 0.707 | 0.583 | 0.639 | 0.670 | 0.694 | 0.682 | 0.676 | |

**Table 10. Pearson (r), Spearman (ρ) and harmonic (h) values obtained in our experiments from the evaluation of ontology similarity methods detailed below in the MedSTS$_{full}$ [52] dataset for each NER tool.**

| ID | Methods | MetaMap | | | MetaMap Lite | | | cTAKES | | |
|----|---------|---------|---|---|--------------|---|---|--------|---|---|
| | | r | ρ | h | r | ρ | h | r | ρ | h |
| M12 | UBSM-Rada | 0.711 | 0.653 | 0.681 | 0.753 | 0.689 | 0.720 | **0.764** | **0.7** | **0.73** |
| M13 | UBSM-J&C | 0.576 | 0.547 | 0.561 | **0.683** | **0.621** | **0.65** | 0.634 | 0.549 | 0.588 |
| M14 | UBSM-cosJ&C | 0.637 | 0.575 | 0.605 | **0.699** | **0.638** | **0.667** | 0.659 | 0.581 | 0.617 |
| M15 | UBSM-coswJ&C | 0.675 | 0.608 | 0.64 | **0.722** | **0.659** | **0.689** | 0.697 | 0.625 | 0.659 |
| M16 | UBSM-Cai | 0.606 | 0.555 | 0.58 | **0.686** | **0.628** | **0.656** | 0.635 | 0.552 | 0.591 |
| M17 | COM | 0.758 | 0.692 | 0.724 | 0.770 | 0.706 | 0.737 | **0.773** | **0.708** | **0.739** |

**Table 11. Harmonic score obtained by each combination of a sentence similarity method with a NER tool in the evaluation of the three sentence similarity datasets.** The p-values shown in this table are obtained by using the method for building uniform size datasets detailed above. The last column shows the p-values corresponding to the t-Student test comparing the performance of each combination with the best pair in each group.

| ID | Methods | NER tool | BIOSSES | MedSTS | CTR | Avg | p-value |
|----|---------|----------|---------|--------|-----|-----|---------|
| | | | h | h | h | h | |
| M12 | UBSM-Rada | cTAKES | 0.800 | 0.730 | 0.785 | 0.772 | — |
| | | MetamapLite | 0.744 | 0.72 | 0.785 | 0.751 | 0.011 |
| | | Metamap | 0.742 | 0.680 | 0.723 | 0.715 | 0.000 |
| M13 | UBSM-J&C | MetamapLite | 0.55 | 0.65 | 0.602 | 0.601 | — |
| | | cTAKES | 0.595 | 0.588 | 0.552 | 0.578 | 0.000 |
| | | Metamap | 0.316 | 0.561 | 0.234 | 0.37 | 0.000 |
| M14 | UBSM-cosJ&C | MetamapLite | 0.631 | 0.667 | 0.674 | 0.657 | — |
| | | cTAKES | 0.681 | 0.617 | 0.626 | 0.641 | 0.002 |
| | | Metamap | 0.537 | 0.605 | 0.434 | 0.525 | 0.000 |
| M15 | UBSM-coswJ&C | cTAKES | 0.749 | 0.659 | 0.693 | 0.700 | — |
| | | MetamapLite | 0.678 | 0.689 | 0.732 | 0.700 | 0.018 |
| | | Metamap | 0.656 | 0.64 | 0.551 | 0.616 | 0.005 |
| M16 | UBSM-Cai | MetamapLite | 0.562 | 0.656 | 0.607 | 0.608 | — |
| | | cTAKES | 0.616 | 0.591 | 0.571 | 0.593 | 0.001 |
| | | Metamap | 0.419 | 0.58 | 0.318 | 0.439 | 0.000 |
| M17 | COM | cTAKES | **0.801** | **0.739** | 0.786 | **0.776** | — |
| | | MetamapLite | 0.788 | 0.737 | **0.789** | 0.772 | 0.052 |
| | | Metamap | 0.792 | 0.724 | 0.768 | 0.761 | 0.004 |

**Table 12. Pearson (r) and Spearman (ρ) correlation values, harmonic score (h), and harmonic average (AVG) score obtained by the LiBlock method in combination with each NER tool using the best pre-processing configuration detailed in Table 7.** In addition, the last column (p-val) shows the p-values for the comparison of the LiBlock method with cTAKES and the remaining NER combinations.

| ID | Sentence similarity methods | BIOSSES [30] | | | MedSTS$_{full}$ [52] | | | CTR [53] | | | AVG | p-val |
|----|----------------------------|--------------|---|---|----------------------|---|---|----------|---|---|-----|-------|
| | | r | ρ | h | r | ρ | h | r | ρ | h | h | |
| M4 | LiBlock-cTAKES | **0.820** | **0.828** | **0.824** | 0.769 | **0.710** | **0.739** | 0.793 | **0.808** | 0.800 | **0.788** | - |
| M4 | LiBlock-noNER | 0.814 | 0.823 | 0.819 | **0.770** | 0.709 | 0.738 | **0.795** | 0.805 | 0.800 | 0.786 | 0.14 |
| M4 | LiBlock-MetamapLite | 0.799 | 0.819 | 0.809 | 0.763 | 0.705 | 0.733 | 0.794 | **0.808** | **0.801** | 0.781 | 0.015 |
| M4 | LiBlock-Metamap | 0.807 | 0.826 | 0.816 | 0.753 | 0.690 | 0.720 | 0.792 | 0.807 | 0.799 | 0.779 | 0.003 |

**Table 13. Raw and pre-processed sentence pairs obtaining the lowest and highest similarity error $E_{sim}$ together with their corresponding Normalized human similarity score (Human) and normalized similarity value (Method) estimated by the LiBlock (M4) method for the raw and pre-processed sentence pairs with the lowest (L) and highest (H) similarity error $E_{sim}$.**

| $E_{sim}$ | Input sentence | Pre-processed sentence analyzed by the method | Human | Method |
|---|---|---|---|---|
| L | s1: "Centrosomes increase both in size and in microtubule-nucleating capacity just before mitotic entry." | s1: "C0242608 increase size C0026046 nucleating capacity mitotic entry" | 0.0 | 0.0 |
| | s2: "Functional studies showed that, when introduced into cell lines, miR-146a was found to promote cell proliferation in cervical cancer cells, which suggests that miR-146a works as an oncogenic miRNA in these cancers." | s2: "functional studies showed introduced C0007634 lines mir 146a found promote C0007634 C0334094 C4048328 C0007634 suggests mir 146a works oncogenic mirna C0006826" | | |
| H | s1: "Consequently miRNAs have been demonstrated to act either as oncogenes (e.g., miR-155, miR-17–5p and miR-21) or tumor suppressors (e.g., miR-34, miR-15a, miR-16–1 and let-7)" | s1: "consequently mirnas demonstrated C0427611 either oncogenes e g mir 155 mir 17 5p mir 21 C0027651 suppressors e g mir 34 mir 15a mir 16 1 let 7" | 0.7 | 0.0 |
| | s2: "Given the extensive involvement of miRNA in physiology, dysregulation of miRNA expression can be associated with cancer pathobiology including oncogenesis], proliferation, epithelial-mesenchymal transition, metastasis, aberrations in metabolism, and angiogenesis, among others" | s2: "given extensive involvement mirna physiology dysregulation mirna C0185117 associated C0006826 pathobiology including oncogenesis C0334094 epithelial mesenchymal transition metastasis aberrations C0025519 angiogenesis among others" | | |

**Table 14. Raw and pre-processed sentence pairs obtaining the lowest and highest similarity error $E_{sim}$ together with their corresponding Normalized human similarity score (Human) and normalized similarity value (Method) estimated by the COM (M17) method for the raw and pre-processed sentence pairs with the lowest (L) and highest (H) similarity error $E_{sim}$. We show the raw and pre-processed sentence pairs evaluated by the WBSM and UBSM similarity methods that make up the COM method. The UBSM method use the cTAKES NER tool.**

| $E_{sim}$ | Input sentence | Pre-processed sentence analyzed by the method | Human | Method |
|---|---|---|---|---|
| Low | s1: "The in vivo data is still preliminary and other potential roadblocks such as drug resistance have not been examined." | s1, WBSM-Rada: "vivo data still preliminary potential roadblocks drug resistance examined"<br>s1, UBSM-Rada: "vivo data still preliminary potential roadblocks C0013227 resistance examined" | 0.0 | 0.0 |
| | s2: "The GEM model used in this study retains wild-type Tp53, suggesting that the tumors successfully treated with bortezomib and fasudil might not be as aggressive as those in most NSCLC patients" | s2, WBSM-Rada: "gem model used study retains wild type tp53 suggesting tumors successfully treated bortezomib fasudil might aggressive nsclc patients"<br>s2, UBSM-Rada: "gem model used study retains wild type tp53 suggesting C0027651 successfully treated C1176309 fasudil might aggressive C0007131 patients" | | |
| High | s1: "The oncogenic activity of mutant Kras appears dependent on functional Craf, but not on Braf" | s1, WBSM-Rada: "oncogenic activity mutant kras appears dependent functional craf braf"<br>s1, UBSM-Rada: "oncogenic C0026606 mutant kras appears dependent functional craf braf" | 0.75 | 0.0 |
| | s2: "Notably, c-Raf has recently been found essential for development of K-Ras-driven NSCLCs" | s2, WBSM-Rada: "notably c raf recently found essential development k ras driven nsclcs"<br>s2, UBSM-Rada: "notably c raf recently found essential development k C0525678 driven nsclcs" | | |

**Table 15. Raw and pre-processed sentence pairs obtaining the lowest and highest similarity error $E_{sim}$ together with their corresponding Normalized human similarity score (Human) and normalized similarity value (Method) estimated by the BioWordVec$_{int}$ (M26) method for the raw and pre-processed sentence pairs with the lowest (L) and highest (H) similarity error $E_{sim}$.**

| $E_{sim}$ | Input sentence | Pre-processed sentence analyzed by the method | Human | Method |
|---|---|---|---|---|
| Low | s1: "The up-regulation of miR-146a was also detected in cervical cancer tissues." | s1: "the up regulation of mir 146a was also detected in cervical cancer tissues" | 1.0 | 0.986 |
| | s2: "The expression of miR-146a has been found to be up-regulated in cervical cancer." | s2: "the expression of mir 146a has been found to be up regulated in cervical cancer" | | |
| High | s1: "This oxidative branch activity is elevated in comparison to many cancer cell lines, where the oxidative branch is typically reduced and accounts for <20% of the carbon flow through PPP." | s1: "this oxidative branch activity is elevated in comparison to many cancer cell lines where the oxidative branch is typically reduced and accounts for < 20% of the carbon flow through ppp" | 0.0 | 0.912 |
| | s2: "The Downward laboratory went all the way from identifying GATA2 as a novel synthetic lethal gene to validating it using Kras-driven GEM models." | s2: "the downward laboratory went all the way from identifying gata2 as a novel synthetic lethal gene to validating it using kras driven gem models" | | |

**Table 16. Raw and pre-processed sentence pairs obtaining the lowest and highest similarity error $E_{sim}$ together with their corresponding Normalized human similarity score (Human) and normalized similarity value (Method) estimated by the OuBioBert (M47) method for the raw and pre-processed sentence pairs with the lowest (L) and highest (H) similarity error $E_{sim}$.**

| $E_{sim}$ | Input sentence | Pre-processed sentence analyzed by the method | Human | Method |
|---|---|---|---|---|
| Low | s1: "Expression of an activated form of Ras proteins can induce senescence in some primary fibroblasts." | s1: "expression activated form ras proteins induce senescence primary fibroblasts" | 0.9 | 0.908 |
| | s2: "The senescent state has been observed to be inducible in certain cultured cells in response to high level expression of genes activated such as the ras oncogene." | s2: "senescent state observed inducible certain cultured cells response high level expression genes activated ras oncogene" | | |
| High | s1: "The in vivo data is still preliminary and other potential roadblocks such as drug resistance have not been examined." | s1: "vivo data still preliminary potential road bl ocks drug resistance examined" | 0.0 | 0.773 |
| | s2: "The GEM model used in this study retains wild-type Tp53, suggesting that the tumors successfully treated with bortezomib and fasudil might not be as aggressive as those in most NSCLC patients" | s2: "gem model used study retains wild type tp53 suggesting tumors successfully treated bortezomib fas udi l might aggressive nsclc patients" | | |

## Comparison of embedding methods

BioWordVec$_{int}$ *(M26) obtains the highest average harmonic score in all datasets among the family of embedding methods detailed in* Table 4*, but does not significantly outperforms all of them. This conclusion can be drawn by looking at the third group of methods in* Table 8 *and checking the p-values reported in Table A.1 in* S1 Appendix*. Table A.1 in* S1 Appendix *shows that the* BioWordVec$_{int}$ *(M26) obtains p-values higher than* $\alpha_c = 0.05/15\ (0,003)$ *when it is compared with the FastText-SkGr-BioC (M33) and Flair (M18) embedding methods.*

BioWordVec$_{int}$ *(M26) obtains the highest Pearson correlation value in the* BIOSSES *and* MedSTS *datasets among the family of embedding methods, whilst the*

**Table 17. Comparison of the mean, minimum and maximum similarity scores of the Normalized Human similarity scores (Human) and the estimated values returned by the best-performing methods of each family in the evaluation of the three biomedical datasets.**

| BIOSSES dataset | | | | |
|---|---|---|---|---|
| ID | Method | Mean similarity | Minimum similarity | Maximum similarity |
| - | Human | 0.549 | 0 | 1 |
| M4 | LiBlock (this work) | 0.194 | 0 | 0.506 |
| M17 | COM [30] | 0.22 | 0 | 0.596 |
| M26 | BioWordVec$_{int}$ [81] | 0.933 | 0.858 | 0.987 |
| M47 | OuBioBert [90] | 0.808 | 0.582 | 0.936 |

| MedSTS dataset | | | | |
|---|---|---|---|---|
| ID | Method | Mean similarity | Minimum similarity | Maximum similarity |
| - | Human | 0.632 | 0 | 1 |
| M4 | LiBlock (this work) | 0.611 | 0 | 1 |
| M17 | COM [30] | 0.631 | 0 | 1 |
| M26 | BioWordVec$_{int}$ [81] | 0.957 | 0.832 | 1 |
| M47 | OuBioBert [90] | 0.885 | 0.437 | 0.997 |

| CTR dataset | | | | |
|---|---|---|---|---|
| ID | Method | Mean similarity | Minimum similarity | Maximum similarity |
| - | Human | 0.254 | 0 | 1 |
| M4 | LiBlock (this work) | 0.103 | 0 | 0.743 |
| M17 | COM [30] | 0.118 | 0 | 0.793 |
| M26 | BioWordVec$_{int}$ [81] | 0.898 | 0.752 | 0.992 |
| M47 | OuBioBert [90] | 0.724 | 0.472 | 0.98 |

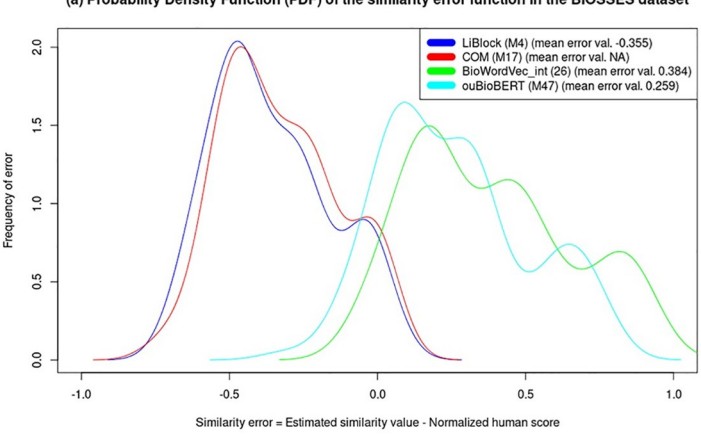

(a) Probability Density Function (PDF) of the similarity error function in the BIOSSES dataset

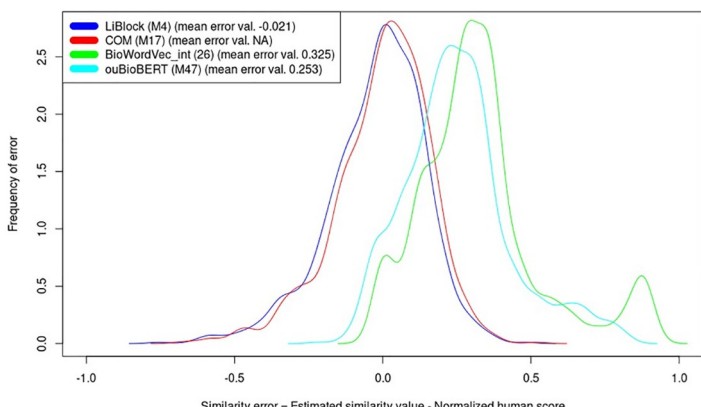

(b) Probability Density Function (PDF) of the similarity error function in the MedSTS dataset

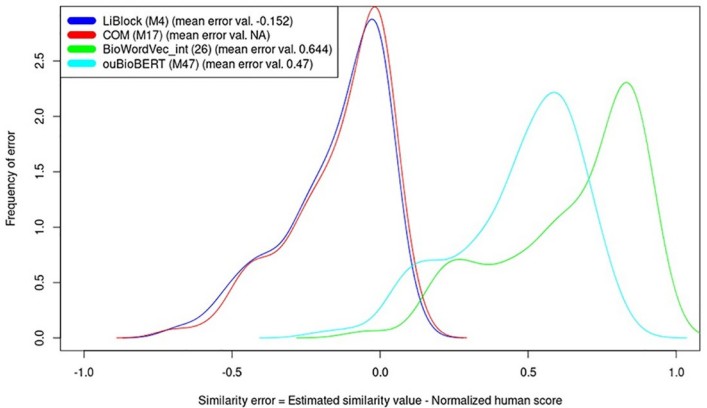

(c) Probability Density Function (PDF) of the similarity error function in the CTR dataset

**Fig 6. Probability Density Function (PDF) and mean value of the similarity error ($E_{sim}$) obtained by the best-performing methods in the evaluation of each dataset as follows: (a) BIOSSES, (b) MedSTS, and (c) CTR.**

*Newman-Griffis*<sub>word2vec_sgns</sub> *(M22) model does so in the CTR dataset.* This conclusion can be drawn by looking at the results for third group of methods detailed in Table 8.

*BioWordVec*<sub>int</sub> *(M26) obtains the highest Spearman correlation in the BIOSSES and MedSTS datasets among the family of embedding methods, whilst the Newman-Griffis*<sub>word2vec_sgns</sub> *(M22)*

*model does so in the CTR dataset. This conclusion can be drawn by looking at the results for the third group of measures detailed in* Table 8.

*BioWordVec*<sub>int</sub> *(M26) obtains the highest harmonic score in the BIOSSES and MedSTS datasets among the family of embedding methods, whilst the Newman-Griffis*<sub>word2vec_sgns</sub> *(M22) model does so in the CTR dataset. This conclusion can be drawn by looking at the results for the third group of measures detailed in* Table 8.

## Comparison of BERT-based methods

*OuBioBERT (M47) obtains the highest average harmonic score among the family of BERT-based methods. However, it does not significantly outperform all of them.* This conclusion can be drawn by looking at the last group of methods in Table 8 and checking the p-values reported in Table A.1 in S1 Appendix. Table A.1 in S1 Appendix shows that ouBioBERT obtains p-values higher than $\alpha_c$ = 0.05/16 (0,003) when it is compared with many BERT-based methods, such as BioBERT Large 1.1 (p-value = 0.224) and PubMedBERT (abstracts+full text) (p-value = 0.101) among others.

*NCBI-BlueBERT Large PubMed (M40) obtains the highest Pearson correlation value in the BIOSSES dataset among the family of BERT-based methods, whilst the NCBI-BlueBERT Base PubMed + MIMIC-III (M41) and the ouBioBERT (M47) models do so in the MedSTS and the CTR datasets, respectively.* This conclusion can be drawn by looking at the last group of measures detailed in Table 8.

*ouBioBERT (M47) obtains the highest Spearman correlation value in the BIOSSES dataset among the family of BERT-based methods, whilst SciBERT (M43) and NCBI-BlueBERT Base PubMed (M39) do so in the MedSTS and CTR datasets, respectively.* These conclusions can be drawn by looking at the last group of measures detailed in Table 8.

*ouBioBERT (M47) obtains the highest harmonic score in the BIOSSES dataset among the family of BERT-based methods, whilst SciBERT (M43) and NCBI-BlueBERT Base PubMed (M39) do so in the MedSTS and CTR datasets, respectively.* This conclusion can be drawn by looking at the last group of measures detailed in Table 8.

## Comparison of all methods

*LiBlock (M4) obtains the highest average harmonic score for all the methods evaluated herein, and significantly outperforms all the methods based on language models. However, there is no a statistically significant difference in performance with the embedding methods Flair (M18) and BioWordVec*<sub>int</sub> *(M26), and the ontology-based methods COM (M17) and WBSM-Rada (M7).* This conclusion can be drawn by looking at the average column in Table 8 and checking the p-values reported in Table A.1 in S1 Appendix. Table A.1 in S1 Appendix shows that the LiBlock obtains p-values higher than $\alpha_c$ = 0.05/16 (0,003) when it is compared with the embedding-based methods Flair (M18) and BioWordVec$_{int}$ (M26). In addition, the LiBlock method obtain p-values higher than $\alpha_c$ = 0.05/11 (0,004) when it is compared with the ontology-based methods COM (M17) and WBSM-Rada (M7).

*BioWordVec*<sub>int</sub> *(M26) obtains the highest Pearson correlation values in the BIOSSES dataset among all methods evaluated here, whilst WBSM-Rada (M7) and Newman-Griffis*<sub>word2vec_sgns</sub> *(M22) do so in the MedSTS and CTR datasets, respectively.* This conclusion can be drawn by looking at the bold values detailed in Table 8.

*LiBlock (M4) obtains the highest Spearman correlation value in the BIOSSES and MedSTS datasets among all methods evaluated here, whilst Newman-Griffis*<sub>word2vec_sgns</sub> *(M22) do so in*

*the CTR dataset.* These conclusions can be drawn by looking at the bold values detailed in Table 8.

*LiBlock (M4) obtains the highest harmonic score in the BIOSSES dataset among all methods evaluated here, whilst WBSM-Rada (M7) and Newman-Griffis<sub>word2vec_sgns</sub> (M22) do so in the MedSTS and CTR datasets, respectively.* This conclusion can be drawn by looking at the bold values detailed in Table 8.

*COM (M17) obtains the second highest average harmonic score among all methods evaluated here, and it is able to outperform significantly all the methods based on language models. However, it does not significantly outperforms all the embedding, ontology or string-based methods.* This conclusion can be drawn by looking at the bold values detailed in Table 8 and checking the p-values reported in Table A.1 in S1 Appendix. Table A.1 in S1 Appendix shows that COM obtains p-values lower than $\alpha_c = 0.05/17$ (0,002) when it is compared with all the methods based on language models. On the other hand, the COM method obtains p-values higher than $\alpha_c = 0.05/6$ (0,008), $\alpha_c = 0.05/11$ (0,004) and $\alpha_c = 0.05/16$ (0,003) respectively, when it is compared with string, ontology and embedding-based methods.

## Non ML-based methods versus ML-based ones

*The string-based method LiBlock (M4) obtain a higher average harmonic score than all the embedding-based methods in all datasets. Moreover, it significantly outperforms all methods based on embedding models, with the only exceptions of Flair (M18) and BioWordVec<sub>int</sub> (M26)* This conclusion can be drawn by looking at the average column in Table 8 and checking the p-values reported in Table A.1 in S1 Appendix. Table A.1 in S1 Appendix shows that LiBlock obtains p-values lower than $\alpha_c = 0.05/16$ (0,003) when it is compared with all the embedding-based methods except for the BioWordVec<sub>int</sub> (p-value 0.003) and Flair (p-value 0.027) methods.

*All string-based methods obtain a higher average harmonic score than all the BERT-based methods considering all datasets, with the only exception of the Levenshtein distance (M5). However, string-based methods do not significantly outperform all BERT-based methods.* This conclusion can be drawn by looking at the average column in Table 8 and checking the p-values reported in Table A.1 in S1 Appendix. Table A.1 in S1 Appendix shows that the string-based methods Qgram (M1), Jaccard (M2), Block distance (M3), Levenshtein distance (M5) and Overlap coefficient (M6) obtain p-values higher than $\alpha_c = 0.05/17$ (0,002) when they are compared with all the BERT-based methods.

*The ontology-based methods COM (M17), WBSM-Rada (M7) and UBSM-Rada (M12) obtain a higher average harmonic score than all the embedding-based methods considering all datasets. However, they do not significantly outperform all embedding-based methods.* This conclusion can be drawn by looking at the average column in Table 8 and checking the p-values reported in Table A.1 in S1 Appendix. Table A.1 in S1 Appendix shows that the ontology-based methods COM (M17), WBSM-Rada (M7) and UBSM-Rada (M12) obtain p-values higher than $\alpha_c = 0.05/16$ (0,003) when they are compared with all the embedding-based methods.

*The ontology-based methods UBSM-Rada (M12), WBSM-Rada (M7), COM (M17) and UBSM-coswJ&C (M15) obtain a higher average harmonic score than all the BERT-based methods. Moreover, the ontology-based methods UBSM-Rada (M12), WBSM-Rada (M7), and COM (M17) significantly outperform all the BERT-based methods.* This conclusion can be drawn by looking at the average column in Table 8 and checking the p-values reported in Table A.1 in S1 Appendix. Table A.1 in S1 Appendix shows that the UBSM-Rada (M12), WBSM-Rada

(M7) and COM (M17) obtain p-values lower than $\alpha_c = 0.05/17$ (0,002) when they are compared with all the BERT-based methods.

*All embedding methods obtain a higher average harmonic score than all BERT-based methods, with the only exceptions of Flair (M18), BioConceptVec*$_\text{glove}$ *(M25), BioConceptVec*$_\text{fastText}$ *(M30) and USE (M31).* This conclusion can be drawn by looking at the last column in Table 8.

*BioWordVec*$_\text{int}$ *(M26) obtains a higher average harmonic score than all the BERT-based methods considering all datasets and significantly outperforms all of them, with the only exception of NCBI-BlueBERT Base PubMed + MIMIC-III (M41).* This conclusion can be drawn by looking at the average column in Table 8 and checking the p-values reported in Table A.1 in S1 Appendix. Table A.1 in S1 Appendix shows that the BioWordVec$_{int}$ (M26) method obtains p-values lower than $\alpha_c = 0.05/17$ (0,002) when it is compared with all the BERT-based methods, except for the NCBI-BlueBERT Base PubMed + MIMIC-III (p-value = 0.002).

## Impact of the NER tools on the ontology-based methods

This section analyzes the impact of the NER tools on the performance of the sentence similarity methods, and studies the overall impact of the NER configurations. Table 10 shows the results obtained on the performance of NER tools for the sentence similarity methods evaluated in the MedSTS dataset [52], whilst Table 11 shows the harmonic and average harmonic scores, as well as the p-values which result from comparing the harmonic score of the best-performing NER tool for each ontology-based method in the three datasets with the harmonic scores obtained by the other two NER tools.

*MetamapLite obtains the highest Pearson, Spearman, and harmonic scores for the MedSTS dataset in combination with UBSM-J&C (M13), UBSM-cosJ&C (M14), UBSM-coswJ&C (M15) and UBSM-Cai (M16), whilst cTAKES obtains the highest Pearson, Spearman and harmonic scores for the MedSTS dataset in combination with UBSM-Rada (M12) and COM (M17).* This latter conclusion can be drawn by looking at the results shown in Table 10.

*cTAKES obtains the highest average harmonic score for the three datasets in combination with UBSM-Rada (M12), UBSM-coswJ&C (M15) and COM (M17) methods, whilst MetamapLite obtains the highest average harmonic score for the three datasets in combination with UBSM-J&C (M13), UBSM-cosJ&C (M14) and UBSM-Cai (M16).* This conclusion can be drawn by looking at the harmonic scores of the NER tools in Table 11.

*cTAKES combined with COM (M17) obtains the best-performing results of ontology-based methods for the three datasets.* This conclusion can be drawn by looking at the average harmonic scores column shown in Table 11.

*cTAKES is the best-performing tool in combination with the UBSM-Rada (M12), UBSM-coswJ&C (M15), and COM (M17) methods in the three datasets, and significantly outperforms MetamapLite and Metamap or the two former methods. However, there is no a statistically significant difference regarding the Metamap tools when it is combined with the COM (M17) method.* This conclusion can be drawn by looking at the average harmonic scores and p-values shown in Table 11, which are lower than $\alpha_c = 0.05/2$ (0,025).

*MetamapLite is the best-performing tool in combination with the UBSM-J&C (M13), UBSM-cosJ&C (M14), and UBSM-Cai (M16) methods in the three datasets, and significantly outperforms cTAKES and Metamap.* This conclusion can be drawn by looking at the average harmonic scores and p-values shown in Table 11, which are lower than $\alpha_c = 0.05/2$ (0,025).

*The choice of the best NER tool for each method significantly impacts their performance in most cases.* This conclusion follows from the conclusions above.

**Answering RQ3.** Our results show that the ontology-based methods obtain their best performance in the task of biomedical sentence similarity when they use either MetamapLite or

cTAKES. Thus, Metamap should not be used in combination with any of the ontology-based methods evaluated here in this task. Likewise, the results and p-values reported Table 11 show that there is a significant difference in the performance of each ontology-based method according to the NER tool used in most cases. The conclusions above confirm that the selection of the NER tool significantly impacts the performance of the sentence similarity methods using it.

## Impact of the NER tools on the new LiBlock measure

This section analyzes the impact of the NER tools on the new $sim_{LiBk}$ similarity measure. Table 12 shows the results obtained by the $sim_{LiBk}$ measure in the three biomedical datasets using its best pre-processing configuration, and annotating the sentences with all the combinations of NER tools. In addition, the aforementioned table details the p-values resulting from comparing the best-performing LiBlock-NER combination with the combinations based on the other two NER tools.

*LiBlock-cTAKES obtains the highest average harmonic score for the three datasets among the LiBlock-NER combinations. However, it does not significantly outperform LiBlock with no use of a NER tool.* This conclusion can be drawn by looking at the average column in Table 12 and checking the p-values in the last column. This conclusion is especially relevant because it shows that there is no statistically significant difference between using a NER tool like cTAKES or not using it, in the case of the LiBlock measure. We conjecture that this result could have two explanations: firstly, the inability of LiBlock to capture semantic relationships beyond the synonymy, and secondly, the current limitations of cTAKES in recognizing all mentions of biomedical entities.

*LiBlock-cTAKES obtains the highest Pearson correlation value in the BIOSSES dataset among all LiBlock-NER combinations, whilst LiBlock with no use of a NER tool obtains the highest Pearson correlation value in the MedSTS and CTR datasets, respectively.* This conclusion can be drawn by looking at the results detailed in Table 12.

*LiBlock-cTAKES obtains the highest Spearman correlation value in the BIOSSES and MedSTS datasets among the LiBlock-NER combinations, whilst LiBlock-cTAKES and LiBlock-MetamapLite obtain the highest Spearman correlation value in the CTR dataset.* This conclusion can be drawn by looking at the results detailed in Table 12.

*LiBlock-cTAKES obtains the highest harmonic correlation value in the BIOSSES and MedSTS datasets among the LiBlock-NER combinations, whilst LiBlock-MetamapLite obtains the highest harmonic correlation value in the CTR dataset.* This conclusion can be drawn by looking at the results detailed in Table 12.

## Impact of the remaining pre-processing stages

This section analyzes the impact of each pre-processing step on the performance of the sentence similarity methods, except for the NER tools already analyzed in the previous section. Finally, we study the overall impact of the pre-processing configurations.

**Impact of tokenization.** *The family of string-based methods obtains its best-performing results either by splitting the sentence on the spaces between words or using the Stanford CoreNLP tokenizer.* This conclusion can be drawn by looking at Table 7, which summarizes the pre-processing tables detailed in S2 Appendix.

*The family of ontology-based methods obtains its best-performing results in combination with the Stanford CoreNLP tokenizer.* This conclusion can be drawn by looking at Table 7.

*The family of methods based on embedding obtains its best-performing results in combination with the Stanford CoreNLP tokenizer, with the only exception of Flair (M18).* This conclusion can be drawn by looking at Table 7.

*No method based on strings, ontologies, or embedding obtains its best-performing results in combination with the BioCNLPTokenizer.* This conclusion can be drawn by looking at Table 7. Thus, the BioCNLPTokenizer should not be used in combination with any method in the abovementioned families in the task of biomedical sentence similarity. On the other hand, we recall that all BERT-based methods evaluated herein can only be used in combination with the WordPiece Tokenizer [91] based on a subword segmentation algorithm, because it is required by the current BERT implementations.

*All families of methods show a strong preference for a specific tokenizer, with the only exception of the string-based one.* This conclusion can be drawn from previous conclusions that confirm the preference of the methods based on ontologies and embedding for the CoreNLP tokenizer, and the mandatory use of the WordPiece tokenizer by the family of BERT-based methods.

**Impact of character filtering.** *The family of string-based methods obtains its best-performing results by using either the BIOSSES char-filtering method or the default method which removes the punctuation marks and special symbols from the sentences, with the only exception of the Levenshtein distance method (M5), which does not remove special characters.* This conclusion can be drawn by looking at Table 7, which summarizes the pre-processing tables detailed in S2 Appendix.

*All ontology-based methods obtain their best-performing results in combination with the BIOSSES char-filtering method.* This conclusion can be drawn by looking at Table 7.

*Most embedding methods obtain their best-performing results in combination with the default char filtering method. However, Flair (M18), BioWordVec (M26,M27), and BioSentVec (M32) do better with BIOSSES char-filtering.* This conclusion can be drawn by looking at Table 7.

*The BERT-based methods do not show a noticeable preference pattern for a specific char filtering method, obtaining their best-performing results with the BIOSSES, Blagec2019, or the default one.* This conclusion can be drawn by looking at Table 7.

**Impact of stop-words removal.** *All string-based methods obtain their best-performing results in combination with the NLTK2018 stop-word list, with the only exception of the Levenshtein distance (M5).* This conclusion can be drawn by looking at Table 7, which summarizes the pre-processing tables detailed in S2 Appendix.

*All ontology-based methods obtain their best-performing results in combination with the NLTK2018 stop-word list, with the only exception of WBSM-J&C (M8), WBSM-cosJ&C (M9), which do not remove stop words.* This conclusion can be drawn by looking at Table 7.

*The methods based on embedding do not show a noticeable preference pattern for a specific stop-word list, obtaining their best-performing results by using the stop-word list of BIOSSES, NLTK2018, or none at all.* This conclusion can be drawn by looking at Table 7.

*The methods based on language models do not show a noticeable preference pattern for a specific stop-word list, obtaining their best-performing results by using the stop-word list of BIOSSES, NLTK2018, or none at all.* This conclusion can be drawn by looking at Table 7.

*The best-performing results for the methods based on strings or ontologies show a noticeable preference for the use of the stop-words list NLTK2018.* This conclusion can be drawn by looking at the Table 7.

**Impact of lower-casing.** *Only 10 of the 50 methods evaluated in this work obtain their best performance without converting words to lowercase at the sentence pre-processing stage.* This conclusion can be drawn by looking at Tables 7 and 8, and the pre-processing tables detailed in S2 Appendix. Moreover, these ten aforementioned methods obtain a low performance in

our experiments, with the sole exception of the BioNLP2016$_{win30}$ (M29) pre-trained model, which obtains the third best Spearman correlation value in the CTR dataset. Thus, our experiments confirm that the lower-casing normalization of the sentences positively impacts the performance of the methods, and it should be considered as the default option in any biomedical sentence similarity task.

We conjecture that lower-casing improves the performance of the families of string-based and ontology-based methods because it improves the exact comparison of words. On the other hand, we also conjecture that the impact of lower-casing the sentences on the families of methods based on embedding and language models strongly depends on the pre-processing methods used in their training.

**Overall impact of pre-processing.** To study the overall impact of the pre-processing stage on the performance of the sentence similarity methods, we selected the configuration reporting the highest (best) and lowest (worst) average harmonic score values for each method, as shown in Table 9. These configurations were selected from a total of 1081 pre-processing configurations reported in S2 Appendix.

*The best-performing methods of each family show a statistically significant difference in performance between their best and worst pre-processing configurations.* This conclusion can be drawn by looking at the average (AVG) and the p-values in Table 9.

**Answering RQ4.** Our results and the conclusions above show that the pre-processing configurations significantly impact the performance of the sentence similarity methods, and thus, they should be specifically defined for each method. All families of methods show a strong preference for a specific tokenizer, with the sole exception of the string-based one. In addition, the BioCNLPTokenizer does not contribute to the best-performing configuration of any method evaluated here. The family of string-based methods shows a preference pattern for using either the BIOSSES or default char filtering method, whilst all ontology-based methods use the BIOSSES char filtering method, and most embedding methods use the default char filtering method. However, BERT-based methods do not show a noticeable preference pattern for a specific char filtering method. On the other hand, the families of string and ontology-based methods show a noticeable preference pattern for the use of the NLTK2018 stop-words list, whilst the families of embedding- and BERT-based methods do not show a noticeable pattern. Finally, the experiments confirm that the lower-casing normalization of the sentences positively impacts the performance of the methods, and it should be considered as the default option in any biomedical sentence similarity task.

## The new state of the art

We establish the new state of the art to answer our RQ1 and RQ2 questions as follows.

The LiBlock (M4) method sets the new state of the art for the sentence similarity task in the biomedical domain (see Table 8), being the best overall performing method to tackle this task. Moreover, LiBlock significantly outperforms all the methods based on language models. However, LiBlock cannot significantly outperform the ontology-based methods COM (M17) and WBSM-Rada (M7), and the embedding-based methods Flair (M18) and BioWordVec$_{int}$ (M26) (see S1 Appendix). Thus, LiBlock is a convincing but non-definitive winner among the biomedical sentence similarity methods evaluated here.

The COM (M17) method sets the new state of the art among the family of ontology-based methods for biomedical sentence similarity, being the best-performing method in this task (see Table 8). Moreover, COM significantly outperforms all methods based on language models (see S1 Appendix).

BioWordVec$_{int}$ (M26) sets the new state of the art among the family of methods based on pre-trained embedding models, being the best-performing method in this task (see Table 8). However, BioWordVec$_{int}$ does not significantly outperforms the remaining methods in the same family (see S1 Appendix).

OuBioBERT (M47) sets the new state of the art among the family of methods based on pre-trained BERT models, being the best-performing method in this task (see Table 8). However, OuBioBERT is unable to outperform significantly all remaining methods from the same family (see S1 Appendix).

Finally, our results show that our new string-based method, called LiBlock (M4), obtains the best overall results, despite not capturing the semantic information of the sentences. This is a very notable finding because it contradicts a common belief that ontology-based methods, which integrate word and concept semantics, will outperform the non-semantic methods in this similarity task. A second and very interesting finding is that our non-semantic and non-ML LiBlock method is able to outperform significantly state-of-the-art methods based on BERT language models [86] in an unsupervised context. This latter finding is very remarkable because LiBlock is easy to implement and evaluate, very efficient (2635 sentence pairs per second with no use of a NER tool), and it requires neither large text resources nor complex algorithms for its training and evaluation, which is a very clear advantage in the biomedical sentence similarity task.

**Answering RQ1 and RQ2.** The string-based method LiBlock (M4) obtains the highest average harmonic score in all datasets, and significantly outperforms the remaining string-based methods, as well as all methods based on language models, and all the ontology-based methods with the only exceptions of COM (M17) and WBSM-Rada (M7). In addition, LiBlock obtains the highest Spearman correlation values in the BIOSSES and MedSTS datasets, which contain 100 and 1068 sentence pairs respectively.

## Main drawbacks and limitations of current methods

This section analyzes the behaviour of the best-performing methods in each family of sentence similarity methods to answer our RQ5. The best-performing methods of each family, according to the harmonic average value reported in Table 8, are LiBlock (M4), COM (M17), BioWordVec$_{int}$ (M26), and OuBioBERT (M47).

*String and ontology-based methods underestimate, on average, the human similarity value in the BIOSSES and CTR datasets, whilst their average similarity error is close to 0 in the MedSTS dataset.* This conclusion can be drawn by looking at the average similarity error values and the mean error values shown in Fig 6 together with the mean values shown in Table 17. LiBlock and COM obtain mean error values of -0.021 and -0.001 in MedSTS, as shown in Fig 6b. On the other hand, both methods report a mean similarity score much lower than the mean of the Human normalized score in the BIOSSES and CTR datasets and a mean similarity score close to the Human normalized score in the MedSTS dataset, as shown in Table 17.

*The methods based on embedding and language models overestimate, on average, the human similarity value in the three datasets.* This conclusion can be drawn by looking at the average similarity error values and the mean error values shown in Fig 6, together with the mean similarity values shown in Table 17. The two aforementioned families of methods report a mean similarity score much higher than the mean of the Human normalized score in the three datasets, as show in Table 17.

*String and ontology-based methods share a similar underestimation behavior, in contrast to the overestimation behaviour shown by the methods based on embedding and language models, which is very noticeable in the three datasets.* This conclusion can be drawn by looking at the

minimum and maximum similarity values columns in Table 17, and the plots of the probability error distribution function for the three datasets in Fig 6. For instance, in spite of the human similarity scores being in the range of 0 to 1 in the BIOSSES dataset, as shown in Table 17, the string and ontology-based methods report similarity scores in the range of 0 to 0.596, whilst the methods based on embedding and language models report similarity scores in the range of 0.582 to 0.987.

*String and ontology-based methods tend to obtain their best results in sentences with a Human normalized score close to 0, whilst the methods based on embedding and language models obtain their best results in sentences with a Human normalized score close to 1.* This conclusion can be drawn by looking at Tables 13–16. On the other hand, string and ontology-based methods tend to obtain their worst results in sentences with a Human normalized score close to 1, whilst the methods based on embedding and language models obtain their worst results in sentences with a Human normalized score close to 0.

*None of the methods for semantic similarity of sentences in the biomedical domain evaluated here use an explicit syntactic analysis or syntax information to obtain the similarity value.* We conjecture that syntactic analysis would improve the performance in some cases. For instance, the sentences *s*1 and *s*2 with highest $E_{sim}$ in Table 13 show an implicit relation between the concepts "miRNA" and "oncogenesis", which should increase the final semantic similarity score of the sentences. However, none of the methods evaluated here consider and reward these semantic relationships because its recognition demands some form of syntactic analysis. On the one hand, string and ontology-based methods consider the concepts in a sentence as bags of words, whilst on the other hand the methods based on embedding and language models implicitly consider the structure of the sentences but not the relationships between the parts of the sentences that are related.

*Our results show that the family of string-based methods benefits from a high frequency of overlapping words in the sentences of the current biomedical datasets, whilst such methods are not able to deal properly with sentences that are semantically different but not exhibit a word overlapping pattern.* The main advantages of the string-based methods are as follows: (1) they are able to obtain high correlation values without the need of using external resources for their training or evaluation; (2) they are fast and efficient; and finally; (3) they require low computational resources. However, string-based methods are unable to capture the semantics of the words in the sentence, which prevent them from recognizing semantic relationships, such as synonymy, meronymy and morphological variants. On the other hand, the use of NER tools in combination with string-based methods is a good option to integrate at least the capability of recognizing synonyms, as shown by LiBlocK-cTAKES (M4).

*Ontology-based methods strongly depend on the lexical coverage of the ontologies and the ability to recognize automatically the underlying concepts in sentences.* Our results show that the ontology-based methods are able to properly estimate a similarity score when used either with a dataset with high word overlapping, or with NER and WSD tools that find all possible entities to properly calculate the similarity between sentences. The main advantages of ontology-based methods are that they are fast and require low computational resources. However, the effectiveness of the ontology-based methods depends on the lexical coverage of the ontologies and the ability of the NER and WSD tools to recognize the underlying concepts in sentences, whose coverage and performance could be limited in several application domains.

The LiBlock (M4) string-based method and the COM (M17) ontology-based method use a NER tool in the pre-processing stage to recognize the biomedical entities (UMLS CUI codes) present in the input sentences. The objective of annotating entities in the semantic similarity task is the identification and disambiguation of biomedical concepts to provide semantic information to sentences. LiBlock uses the NER tool to normalize and disambiguate the underlying

concepts in a sentence, unifying different concepts with acronyms and synonyms in the same CUI code and creating an overlapping between concepts, while ontologies also make use of the similarity of concepts within ontologies.

*The biomedical NER tools evaluated in this work are unable to identify and disambiguate correctly many biomedical concepts due to the use of acronyms and different morphological variations, among others.* For example, the CUI concepts "KRAS gene" (C1537502), "BRAF gene" (C0812241), and "RAF1 gene" (C0812215) in the sentences $s1$ and $s2$ with highest $E_{sim}$ obtained by the COM (M17) method in Table 14, appear as "K-ras", "Braf", "c-Raf" and "Craf". However, cTAKES is unable to recognize these later morphological variants of the same biomedical concepts. A second example is the word "act" in the sentence "Consequently miRNAs have been demonstrated to act either as oncogenes [. . .]", which is wrongly recognized as the entity "Activated clotting time measurement" (C0427611), rather than as a verb in the sentence $s1$ with highest $E_{sim}$ in Table 13. And finally, a third example is the acronym "NSCLC", which denotes the concept "Non-Small Cell Lung Carcinoma (C0007131), which is not recognized in the plural variant "NSCLCs" in the sentence $s2$ with highest $E_{sim}$ from Table 14.

The methods based on pre-trained embedding and language models provide a broader lexical coverage than the ontology-based methods, and do not need the use of NER or WSD tools to find intrinsic semantic relationships between the words in the sentences. However, these methods need large corpora for their training, as well as a complex training phase and more computational resources than the methods from the string-based and ontology-based families. Moreover, our experiments show that those methods tend to estimate higher similarity values than those estimated by a human being in the three datasets. In most cases, the aforementioned methods report similarity scores that tend towards 1, which indicates that the semantics obtained from the sentences is not sufficient to compute correctly a similarity score. For instance, the sentences $s1$ and $s2$ with highest $E_{sim}$ from Tables 15 and 16 shows similarity values close to 1, where the sentences have neither word overlapping nor similar concepts, and the human similarity score is 0 in both cases. Lastly, BERT-based methods, are trained for downstream tasks, using a supervised approach, and do not perform well in an unsupervised context.

**Answering RQ5.** String-based methods capture neither the word semantics within the sentences nor the semantic relationships between words, such as synonymy and meronymy, and their effectiveness mainly relies on the word overlapping frequency in the sentences. However, the LiBlock method uses the NER tool to normalize and disambiguate the underlying concepts in a sentence, but unfortunately, it does not significantly outperform LiBlock with no use of a NER tool, which could have two explanations: firstly, the inability of LiBlock to capture semantic relationships beyond the synonymy; secondly, the current limitations of cTAKES in recognizing all mentions of biomedical entities. On the other hand, ontology-based methods use NER and WSD tools to recognize the underlying concepts in the sentences, which are not able correctly to identify and disambiguate these concepts in many cases. In addition, they require external resources to capture the semantic information from the sentences, which limits their lexical coverage. Thus, ontology-based methods require both high word overlapping and high recognition coverage of named entities to properly estimate the similarity between sentences. In comparison, the methods based on pre-trained embedding and language models need large corpora for training, a complex training phase, and considerable computational resources to calculate the similarity between sentences. Moreover, those methods tend to obtain high similarity scores in most cases, which may penalize them in a balanced dataset and in a real environment. Finally, BERT-based methods are trained for downstream tasks, using a supervised approach, and do not perform well in an unsupervised context.

**Table 18. This table shows the running times in milliseconds (ms) and the average sentences pairs per second (sent/sec) reported by the best-performing method of each family of methods in the evaluation of the 1339 sentence pairs that comprise the three datasets.** (*) The LiBlock method reports the running times in both NER and noNER versions showing that the efficiency of the method with no NER tool is much higher, despite the fact that there is no statistically significant difference in the results between both pre-processing configurations.

| ID | Method | Running time (ms) | Sentence pairs / sec |
|----|--------|-------------------|----------------------|
| M4 | LiBlock-cTAKES | 56605 | 23,66 |
| M4 | LiBlock-noNER (*) | 508 | 2635,83 |
| M3 | Block distance | 308 | **4347,4** |
| M12 | UBSM-Rada | 32341 | 41,40 |
| M17 | COM | 41558 | 32,22 |
| M27 | BioWordVec$_{int}$ | 1211 | 1105,69 |
| M32 | BioSentVec | 54706 | 24,48 |
| M47 | ouBioBERT | 575770 | 2,33 |
| M38 | BioBERT Large 1.1 (+ PubMed) | 3312566 | 0,40 |

## Comparison of running times

Table 18 details the running time reported by the best-performing methods for each family, as well as the sentences per second that each method computes on average for the three datasets evaluated herein. The experiments were executed on a desktop computer with an AMD Ryzen 7 5800x CPU (16 cores) with 64 Gb RAM and a 2TB Gb SSD disk. In all cases, the running time includes the pre-processing time for each method. The string-based method Block Distance (M3) obtains the lowest running times because it does not need complex mechanisms or pre-trained models to calculate the similarity between sentences. On the other hand, the BERT-based methods obtain the worst results mainly due to their pre-processing stage, which uses the WordPiece tokenization method.

## Inconsistent results in the calculation of the statistical significance matrix

Despite the artificial increase of datasets to calculate the statistical significance of the results, we have identified an inconsistent result with respect to the comparison of the p-values of the LiBlock (M4) and the WBSM-Rada (M7) and UBSM-Rada (M12) methods. Table 8 shows that the UBSM-Rada method (M12) has a higher average harmonic score compared to WBSM-Rada (M7). However, by building the artificial datasets, the value of UBSM-Rada (M12) with respect to LiBlock (M4) shows a significant difference, while WBSM-Rada (M7) with respect to LiBlock (M4) shows a non-significant difference. We conjecture that this problem could be solved by increasing the number of datasets created for this task, which would allow the sample size to be increased and obtain more consistent results.

## Conclusions and future work

We have introduced the largest, detailed, and for the first time, reproducible experimental survey on biomedical sentence similarity reported in the literature. Our work also introduces a collection of self-contained and reproducible benchmarks on biomedical sentence similarity based on the same software platform, called HESML-STS, which has been especially developed for this work, being provided as part of the new HESML V2R1 version that is publicly available [105]. We provide a detailed reproducibility protocol [44] and dataset [43] to allow the exact replication of all our experiments, methods, and results. In addition, we introduce a new aggregated string-based sentence similarity method called LiBlock, together with eight variants

of the ontology-based methods introduced by Sogancioglu et al. [30], and a new pre-trained word embedding model based on FastText [58] and trained on the full-text of the articles in the PMC-BioC corpus [19]. We also evaluate for the first time the CTR [53] dataset in a benchmark on biomedical sentence similarity.

The string-based LiBlock (M4) measure sets the new state of the art for the sentence similarity task in the biomedical domain and significantly outperforms all the methods of each family evaluated here, with the only exceptions of the Flair (M18), BioWordVec$_{int}$ (M26), COM (M17) and WBSM-Rada (M7) methods. However, our data analysis shows that at least with the three datasets evaluated herein, there is no statistically significant difference between the performance of the LiBlock (M4) method using the cTAKES or using no NER tool at all. Thus, using the LiBlock method without any NER tool could be a competitive and much more efficient solution for high-throughput applications.

Concerning the impact of the Named Entity Recognition (NER) tools, our results confirm that the choice of the best NER tool for each method significantly impacts their performance. MetamapLite [94] and cTAKES [62] set the best-performing configurations for the family of ontology-based methods, whilst Metamap [34] was not the best performer in any method.

Our experiments confirm that the pre-processing stage has a very significant impact on the performance of the sentence similarity methods evaluated here, and yet this aspect has neither been studied nor reported in the literature. Thus, the selection of the proper configuration for each sentence similarity method should be confirmed experimentally. However, our experiments suggest some default configurations to make these decisions, such as the use of lower-casing normalization, some specific char filtering methods, and some specific tokenizers with the sole exception of BioCNLPTokenizer. Finally, the families of string and ontology-based methods show a noticeable preference pattern for the use of the NLTK2018 stop-words list. For a detailed description of the best pre-processing configurations, we refer the readers to our discussion.

String-based methods do not capture either the semantics of the words in the sentence or the semantic relationships between words, and their effectiveness relies on the word overlapping frequency in the sentences. Ontology-based methods Named Entity Recognition (NER) and Word Sense Disambiguation (WSD) tools to recognize the underlying concepts in the sentences and require external resources to capture the semantic information from the sentences, which limits their lexical coverage. In addition, they require either high word overlapping or high recognition coverage of named entities in order to properly calculate the similarity between sentences. On the other hand, the methods based on pre-trained embedding and language models need a large corpus for training, a complex training phase, and considerable computational resources to calculate the similarity between sentences. Moreover, these methods tend to obtain high similarity scores in most cases, which may penalize them in a balanced dataset and in a real environment. Finally, BERT-based methods are trained for downstream tasks, using a supervised approach, and do not perform well in an unsupervised context.

Our experiments suggest that the current benchmarks do not cover all the language features that characterize the biomedical domain, such as the frequent use of acronyms and rhetorical expressions like synonymy, meronymy, etc. In addition, current benchmarks have a very limited sample size that make the analysis of results difficult. We conjecture that LiBlock, COM, and UBSM-Rada perform well because there is a noticeable overlap of terms that may benefit these methods over the others reported in the literature. Furthermore, Chen et al. [106] highlight the need to improve and create new benchmarks from different perspectives, to reflect the multifaceted notion of the similarity of sentences. Therefore, we found a strong need for improving existing benchmarks for the task of semantic similarity of sentences in the biomedical domain.

As part of our forthcoming activities, we plan to evaluate the new sentence similarity methods introduced herein in a benchmark for the general language domain. In addition, we will study the evaluation of sentence similarity methods in an extrinsic task, such as semantic medical indexing [107] or summarization [108]. We also consider the evaluation of further pre-processing configurations, such as biomedical NER systems based on recent Deep Learning techniques [10], or extending our experiments and research to the multilingual scenario by integrating multilingual biomedical NER systems like Cimind [109]. Finally, we plan to evaluate some recent biomedical concept embeddings based on MeSH [35], which has not been evaluated in the sentence similarity task yet.

## Supporting information

**S1 Appendix. The statistical significance results.** We provide a series of tables reporting the p-values for each pair of methods evaluated in this work as supplementary material. (PDF)

**S2 Appendix. The pre-processing raw output files.** We provide all the pre-processing raw output tables for the experiments evaluated herein as supplementary material. (PDF)

**S3 Appendix. A reproducibility protocol and dataset on the biomedical sentence similarity.** We provide the reproducibility protocol published at protocols.io [44] as supplementary material to allow the exact replication of all our experiments, methods, and results. (PDF)

## Acknowledgments

We are grateful to Gizem Sogancioglu and Kathrin Blagec for answering kindly our questions to replicate their methods and experiments, Fernando González and Juan Corrales for setting up our reproducibility dataset, and Hongfang Liu and Yanshan Wang for providing us the MedSTS dataset. UMLS CUI codes, SNOMED-CT US ontology and MeSH thesaurus were used in our experiments by courtesy of the National Library of Medicine of the United States. Finally, we thank David Pritchard for checking the use of English in our manuscript.

## Author Contributions

**Conceptualization:** Alicia Lara-Clares, Juan J. Lastra-Díaz.

**Data curation:** Alicia Lara-Clares.

**Formal analysis:** Alicia Lara-Clares, Juan J. Lastra-Díaz.

**Funding acquisition:** Ana Garcia-Serrano.

**Investigation:** Alicia Lara-Clares, Juan J. Lastra-Díaz.

**Methodology:** Alicia Lara-Clares, Juan J. Lastra-Díaz.

**Resources:** Alicia Lara-Clares.

**Software:** Alicia Lara-Clares, Juan J. Lastra-Díaz.

**Supervision:** Juan J. Lastra-Díaz, Ana Garcia-Serrano.

**Validation:** Alicia Lara-Clares, Juan J. Lastra-Díaz.

**Visualization:** Alicia Lara-Clares.

**Writing – original draft:** Alicia Lara-Clares.

**Writing – review & editing:** Alicia Lara-Clares, Juan J. Lastra-Díaz.

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
