## [Decision Letter · Decision Letter 0]

18 Jul 2022

PONE-D-22-05195A reproducible experimental survey on biomedical sentence similarity: a string-based method sets the state of the artPLOS ONE

Dear Dr. Lara-Clares,

Thank you for submitting your manuscript to PLOS ONE. After careful consideration, we feel that it has merit but does not fully meet PLOS ONE’s publication criteria as it currently stands. Therefore, we invite you to submit a revised version of the manuscript that addresses the points raised during the review process.

ACADEMIC EDITOR: Authors are requested to address the comments of the reviewer. 

Furthermore, the manuscript should be refined for English grammatical structure and phraseology. The manuscript should be polished by an English linguist or language service (note in marked-up copy text where changes are made). 'Grammarly' is poor quality and not acceptable.

We look forward to receiving your revised manuscript.

Kind regards,

Usman Qamar

Academic Editor

PLOS ONE

Journal Requirements:

“This work was partially supported by the UNED predoctoral grant started in April 2019 (BICI N7, November 19th, 2018) and the CLARA-HD (PID2020-116001RB-C32) project.

Reviewers' comments:

Reviewer's Responses to Questions

**Comments to the Author**

1. Does the manuscript adhere to the experimental procedures and analyses described in the Registered Report Protocol?

If the manuscript reports any deviations from the planned experimental procedures and analyses, those must be reasonable and adequately justified.

Reviewer #1: Yes

2. If the manuscript reports exploratory analyses or experimental procedures not outlined in the original Registered Report Protocol, are these reasonable, justified and methodologically sound?

A Registered Report may include valid exploratory analyses not previously outlined in the Registered Report Protocol, as long as they are described as such.

Reviewer #1: Yes

3. Are the conclusions supported by the data and do they address the research question presented in the Registered Report Protocol?

The manuscript must describe a technically sound piece of scientific research with data that supports the conclusions. The conclusions must be drawn appropriately based on the research question(s) outlined in the Registered Report Protocol and on the data presented.

Reviewer #1: Yes

4. Have the authors made all data underlying the findings in their manuscript fully available?

Reviewer #1: Yes

5. Is the manuscript presented in an intelligible fashion and written in standard English?

Reviewer #1: Yes

6. Review Comments to the Author

Please use the space provided to explain your answers to the questions above. (Please upload your review as an attachment if it exceeds 20,000 characters)

Reviewer #1: The authors present a very extensive and detailed analysis of algorithms for computing semantic similarity of sentences from the biomedical literature. The work is relevant and well-executed.

My main source of confusion are the methods used for calculating statistical significance. First, the authors split a dataset into multiple artificially created sub-datasets to enable statistical significance testing. I wonder what the implications of this are, since the datasets so created come from the same data distribution. Please elucidate this a bit more, or point to references that discuss such a procedure from a statistical perspective (or overhaul this part of the analysis if it turns out that statistical significance cannot be calculated as orignally planned.)

Second, it seems like many comparisons are made and no corrections for multiple testing are made.

Please elucidate this and point out why this is not needed (or if it is needed, add corrections for multiple testing).

Minor comments:

Position 33-35: "cannot be reproduced because of the lack of source code and data" [...]:

-> Please check if this strong statement is correct. For example, all code and data used in the study of Blagec et al. is available at https://github.com/kathrinblagec/neural-sentence-embedding-models-for-biomedical-applications (and this link is provided in the manuscript)

7. PLOS authors have the option to publish the peer review history of their article (what does this mean?). If published, this will include your full peer review and any attached files.

Reviewer #1: No

---

## [Author Response · Author response to Decision Letter 0]

29 Sep 2022

We are very grateful for your significant effort to review our manuscript, as well as your kind remarks and suggestions to improve the quality of the paper. 

We have accepted and followed [all] suggestions made by the reviewers. Likewise, the use of English has been revised by a native English-speaker linguist, as suggested by the Editor.

Finally, we provide below our detailed answer for each suggestion made by the reviewers.

Most sincerely,

The authors

Comments:

The authors present a very extensive and detailed analysis of algorithms for computing semantic similarity of sentences from the biomedical literature. The work is relevant and well-executed.

[Authors] Thank you so much for your kind remarks.

My main source of confusion are the methods used for calculating statistical significance. First, the authors split a dataset into multiple artificially created sub-datasets to enable statistical significance testing. I wonder what the implications of this are, since the datasets so created come from the same data distribution. Please elucidate this a bit more, or point to references that discuss such a procedure from a statistical perspective (or overhaul this part of the analysis if it turns out that statistical significance cannot be calculated as orignally planned.)

[Authors] To clarify the issue mentioned above, we have included a detailed explanation in lines 384-413 on page 16 supported by the data and statistical tests shown in the new figure 5. In short, we detail and explain the statistical reasons that encouraged us to split MedSTS into smaller datasets for our statistical significance analysis. We show that the harmonic score used as an evaluation metric is a random variable with different normal distributions depending on the size of the dataset used for its computation.

Second, it seems like many comparisons are made and no corrections for multiple testing are made.

Please elucidate this and point out why this is not needed (or if it is needed, add corrections for multiple testing).

[Authors] We thank the reviewer for pointing out this flaw in our statistical significance analysis. For this reason, we have defined a Bonferroni correction to evaluate the statistical significance of multiple hypothesis tests involved in those conclusions in which one baseline sentence similarity measure is compared with a family of similarity measures, as detailed in lines 414-428 on page 16 of the revised manuscript. All the pairwise p-values keep the same values reported in our previous manuscript. However, we define a Bonferroni corrected null-hypothesis rejection threshold $\\alpha_c$ for the old p-values, such that the null-hypothesis is now rejected only if ``$p-value <= \\alpha_c = 0.05/m$" for each conclusion, where 'm' is the number of comparisons.

Minor comments:

Position 33-35: "cannot be reproduced because of the lack of source code and data" [...]:

-> Please check if this strong statement is correct. For example, all code and data used in the study of Blagec et al. is available at https://github.com/kathrinblagec/neural-sentence-embedding-models-for-biomedical-applications (and this link is provided in the manuscript)

[Authors] Done. We have included a proper explanation in section Introduction as shown in new lines 37-51. We have reviewed and refined the statement, providing more detailed information and links to the resources provided by the authors.

---

## [Decision Letter · Decision Letter 1]

10 Oct 2022

A reproducible experimental survey on biomedical sentence similarity: a string-based method sets the state of the art

PONE-D-22-05195R1

Dear Dr. Lara-Clares,

We’re pleased to inform you that your manuscript has been judged scientifically suitable for publication and will be formally accepted for publication once it meets all outstanding technical requirements.

Kind regards,

Usman Qamar

Academic Editor

PLOS ONE

Additional Editor Comments (optional):

Reviewers' comments:

Reviewer's Responses to Questions

**Comments to the Author**

1. Does the manuscript adhere to the experimental procedures and analyses described in the Registered Report Protocol?

If the manuscript reports any deviations from the planned experimental procedures and analyses, those must be reasonable and adequately justified.

Reviewer #1: Yes

2. If the manuscript reports exploratory analyses or experimental procedures not outlined in the original Registered Report Protocol, are these reasonable, justified and methodologically sound?

A Registered Report may include valid exploratory analyses not previously outlined in the Registered Report Protocol, as long as they are described as such.

Reviewer #1: Yes

3. Are the conclusions supported by the data and do they address the research question presented in the Registered Report Protocol?

The manuscript must describe a technically sound piece of scientific research with data that supports the conclusions. The conclusions must be drawn appropriately based on the research question(s) outlined in the Registered Report Protocol and on the data presented.

Reviewer #1: Yes

4. Have the authors made all data underlying the findings in their manuscript fully available?

Reviewer #1: Yes

5. Is the manuscript presented in an intelligible fashion and written in standard English?

Reviewer #1: Yes

6. Review Comments to the Author

Please use the space provided to explain your answers to the questions above. (Please upload your review as an attachment if it exceeds 20,000 characters)

Reviewer #1: The authors have sufficiently addressed my previous comments regarding the statistical analysis. Overall the paper is done in a very solid and methodological manner.

7. PLOS authors have the option to publish the peer review history of their article (what does this mean?). If published, this will include your full peer review and any attached files.

Reviewer #1: No

---

## [Editor Report · Acceptance letter]

24 Oct 2022

PONE-D-22-05195R1 

A reproducible experimental survey on biomedical sentence similarity: a string-based method sets the state of the art 

Dear Dr. Lara-Clares:

I'm pleased to inform you that your manuscript has been deemed suitable for publication in PLOS ONE. Congratulations! Your manuscript is now with our production department. 

Kind regards, 

on behalf of

Dr. Usman Qamar 

Academic Editor

PLOS ONE